# Condensing Functions and Approximate Endpoint Criterion for the Existence Analysis of Quantum Integro-Difference FBVPs

**Shahram Rezapour** [1,2] , **Atika Imran** [3] , **Azhar Hussain** [3] , **Francisco Martínez** [4,*] , **Sina Etemad** [2] and **Mohammed K. A. Kaabar** [5]

1    Department of Medical Research, China Medical University Hospital, China Medical University, Taichung 406040, Taiwan; sh.rezapour@azaruniv.ac.ir
2    Department of Mathematics, Azarbaijan Shahid Madani University, Tabriz 53751-71379, Iran; sina.etemad@azaruniv.ac.ir
3    Department of Mathematics, University of Sargodha, Sargodha 40100, Pakistan; atikaimran977@gmail.com (A.I.); azhar.hussain@uos.edu.pk (A.H.)
4    Department of Applied Mathematics and Statistics, Technological University of Cartagena, 30203 Cartagena, Spain
5    Department of Mathematics and Statistics, Washington State University, Pullman, WA 99163, USA; mohammed.kaabar@wsu.edu
*    Correspondence: f.martinez@upct.es; Tel.: +34-968-325-586

**Abstract:** A nonlinear quantum boundary value problem (q-FBVP) formulated in the sense of quantum Caputo derivative, with fractional q-integro-difference conditions along with its fractional quantum-difference inclusion q-BVP are investigated in this research. To prove the solutions' existence for these quantum systems, we rely on the notions such as the condensing functions and approximate endpoint criterion (AEPC). Two numerical examples are provided to apply and validate our main results in this research work.

**Keywords:** condensing function; approximate endpoint criterion; quantum integro-difference BVP; existence

**MSC:** 34A08; 34A12

## 1. Introduction

It is a fact supported by many researchers that fractional calculus (FC) establishes a flexible extension for the classical one to arbitrary orders. FC has attracted particular attention from many researchers of mathematics, applied sciences, and engineering because of the various important applications of this field in modeling certain scientific phenomena and complex physical systems. Modeling systems using fractional derivatives can provide a good interpretation of the physical behavior of the studied systems due to the nonlocality and memory effects that have been exhibited in some systems. Some studies have been conducted on the mathematical analysis of FC and its applications such as European option pricing models [1], p-Laplacian nonperiodic nonlinear boundary value problem [2], nonlocal Cauchy problem [3], economic models involving time fractal [4], complex integral [5], incompressible second-grade fluid models [6], complex-valued functions of a real variable [7], and separated homotopy method [8]. Likewise, quantum calculus is a corresponding field of the standard infinitesimal one without the concept of limits. In spite of the long history that they already have, both theories are in the field of mathematical analysis, the investigation of their properties has emerged not so long ago. The quantum fractional calculus (q-fractional calculus), considered as the fractional correspondence of the q-calculus, was initially proposed by Jackson [9–11]. Researchers such as Al-Salam [12] and Agarwal [13] gave a great boost to the fractional q-calculus and obtained important theoretical results. Based on these results, the fractional q-calculus has emerged as an

instrument with great potential in the field of applications [14–17]. Even in recent years, many articles have been appeared on quantum integro-difference boundary value problems (BVPs), which are valuable abstract tools for modeling many phenomena in various fields of science [18–30].

Asawasamrit et al. [31] provided a multi-term q-integro-difference equation subject to nonlocal multi-quantum integral conditions displayed as

$$
\begin{cases}
{}^{R}_{q_1}\mathfrak{D}^{\varsigma}_{0^+}\hbar(r) = \phi(r, \hbar(r), {}^{R}_{q_2}\mathfrak{I}^{\sigma_1}_{0^+}\hbar(r)), & (r \in [0, K]), \\[2mm]
\hbar(0) = 0, \qquad \nu\, {}^{R}_{q_3}\mathfrak{I}^{\sigma_2}_{0^+}\hbar(\eta_1) = {}^{R}_{q_4}\mathfrak{I}^{\sigma_3}_{0^+}\hbar(\eta_2),
\end{cases}
$$

where $q_1, q_2, q_3, q_4 \in (0, 1)$, $\varsigma \in (1, 2)$, $\sigma_1, \sigma_2, \sigma_3 > 0$, $\eta_1, \eta_2 \in (0, K)$ and $\nu \in \mathbb{R}$. The approach implemented by them to arrive at the existence property of solutions for the suggested q-BVP is based on the fixed-point techniques [31]. After that in 2015, Etemad, Ettefagh and Rezapour [32] concerned the three-term q-difference FBVP

$$
({}^{C}_{q}\mathfrak{D}^{\varsigma}_{0^+}\hbar)(r) = w(r, \hbar(r), {}^{C}_{q}\mathfrak{D}^{1}_{0^+}\hbar(r)),
$$

with four-point q-integro-difference conditions

$$
\lambda_1 \hbar(0) + \zeta_1\, {}^{C}_{q}\mathfrak{D}^{1}_{0^+}\hbar(0) = m_1\, {}^{R}_{q}\mathfrak{I}^{\beta}_{0^+}\hbar(\xi_1) = m_1 \int_0^{\xi_1} \frac{(\xi_1 - qv)^{(\beta-1)}}{\Gamma_q(\beta)} \hbar(v)\, d_q v,
$$

$$
\lambda_2 \hbar(1) + \zeta_2\, {}^{C}_{q}\mathfrak{D}^{1}_{0^+}\hbar(1) = m_2\, {}^{R}_{q}\mathfrak{I}^{\beta}_{0^+}\hbar(\xi_2) = m_2 \int_0^{\xi_2} \frac{(\xi_2 - qv)^{(\beta-1)}}{\Gamma_q(\beta)} \hbar(v)\, d_q v,
$$

where $0 \le r \le 1$, $1 < \varsigma \le 2$, $q \in (0, 1)$, $\beta \in (0, 2]$, $\lambda_1, \lambda_2, \zeta_1, \zeta_2, m_1, m_2 \in \mathbb{R}$ and $\xi_1, \xi_2 \in (0, 1)$ with $\xi_1 < \xi_2$. Ntouyas and Samei [33] turned to studying the solutions' existence for the q-integro-difference FBVP

$$
{}^{C}_{q}\mathfrak{D}^{\varsigma}_{0^+}h(r) = w(r, h(r), (\phi_1 h)(r), (\phi_2 h)(r), {}^{C}_{q}\mathfrak{D}^{\varsigma_1}_{0^+}h(r), {}^{C}_{q}\mathfrak{D}^{\varsigma_2}_{0^+}h(r), \ldots, {}^{C}_{q}\mathfrak{D}^{\varsigma_n}_{0^+}h(r)),
$$

via boundary conditions $h(0) + ah(1) = 0$ and $h'(0) + bh'(1) = 0$, in which $r \in [0, 1]$, $q \in (0, 1)$, $1 < \varsigma < 2$, $\varsigma_k \in (0, 1)$ with $k = 1, 2, \ldots, n$, $a, b \neq -1$, $\phi_m$ are defined by the rule $(\phi_m h)(r) = \int_0^r \mu_m(r, v) h(v)\, d_q v$ for $m = 1, 2$ and $w : [0, 1] \times \mathbb{R}^{n+3} \to \mathbb{R}$ is assumed to be continuous with respect to all $(n + 4)$ variables [33].

Stimulated by the above research studies, the following proposed nonlinear Caputo fractional quantum BVP is furnished with the fractional quantum integro-conditions:

$$
\begin{cases}
{}^{C}_{q}\mathfrak{D}^{\varsigma}_{0^+}\hbar(r) = \varphi_*(r, \hbar(r)), & (\varsigma \in (2, 3),\ q \in (0, 1)), \\[2mm]
\hbar(0) + \hbar(\xi) = \ell_1\, {}^{R}_{q}\mathfrak{I}^{\sigma}_{0^+}\hbar(1), & (\ell_1 \in \mathbb{R}^{>0}), \\[2mm]
{}^{C}_{q}\mathfrak{D}^{\varrho}_{0^+}\hbar(0) + {}^{C}_{q}\mathfrak{D}^{\varrho}_{0^+}\hbar(\xi) = \ell_2\, {}^{R}_{q}\mathfrak{I}^{\sigma}_{0^+}\big[{}^{C}_{q}\mathfrak{D}^{\varrho}_{0^+}\hbar\big](1), & (\ell_2 \in \mathbb{R}^{>0}), \\[2mm]
{}^{C}_{q}\mathfrak{D}^{1}_{0^+}\hbar(0) + {}^{C}_{q}\mathfrak{D}^{1}_{0^+}\hbar(\xi) = \ell_3\, {}^{R}_{q}\mathfrak{I}^{\sigma}_{0^+}\big[{}^{C}_{q}\mathfrak{D}^{1}_{0^+}\hbar\big](1), & (\ell_3 \in \mathbb{R}^{>0}),
\end{cases}
\tag{1}
$$

along with its inclusion version given by

$$
\begin{cases}
{}^{C}_{q}\mathfrak{D}^{\varsigma}_{0^+}\hbar(r) \in \mathbb{T}_*(r, \hbar(r)), & (\varsigma \in (2, 3),\ q \in (0, 1)), \\[2mm]
\hbar(0) + \hbar(\xi) = \ell_1\, {}^{R}_{q}\mathfrak{I}^{\sigma}_{0^+}\hbar(1), & (\ell_1 \in \mathbb{R}^{>0}), \\[2mm]
{}^{C}_{q}\mathfrak{D}^{\varrho}_{0^+}\hbar(0) + {}^{C}_{q}\mathfrak{D}^{\varrho}_{0^+}\hbar(\xi) = \ell_2\, {}^{R}_{q}\mathfrak{I}^{\sigma}_{0^+}\big[{}^{C}_{q}\mathfrak{D}^{\varrho}_{0^+}\hbar\big](1), & (\ell_2 \in \mathbb{R}^{>0}), \\[2mm]
{}^{C}_{q}\mathfrak{D}^{1}_{0^+}\hbar(0) + {}^{C}_{q}\mathfrak{D}^{1}_{0^+}\hbar(\xi) = \ell_3\, {}^{R}_{q}\mathfrak{I}^{\sigma}_{0^+}\big[{}^{C}_{q}\mathfrak{D}^{1}_{0^+}\hbar\big](1), & (\ell_3 \in \mathbb{R}^{>0}),
\end{cases}
\tag{2}
$$

where $r \in [0,1]$, $\xi \in (0,1)$, $\varrho \in (1,2)$ and $\sigma > 0$. Two operators ${}^{C}_{q}\mathfrak{D}^{(\cdot)}_{0^+}$ and ${}^{R}_{q}\mathfrak{I}^{(\cdot)}_{0^+}$ represent the Caputo quantum derivative (CpQD) and the Riemann-Liouville quantum integral (RLQI). Furthermore, continuous single-valued function $\varphi_* : [0,1] \times \mathbb{R} \to \mathbb{R}$ and multi-valued function $\mathbb{T}_* : [0,1] \times \mathbb{R} \to \mathbb{P}(\mathbb{R})$ are assumed to be arbitrary equipped with some required specifications that will be explained subsequently. In comparison to other researches on the quantum difference BVPs that were published in the literature, we here deal with two abstract and extended structures of new fractional quantum difference equations/inclusions via q-integro-difference conditions in which the existing property of the relevant solutions is derived by terms of new notions of the functional analysis such as the condensing maps and the measure of noncompactness and the approximate endpoint criterion. These procedures on the suggested q-difference-BVPs (1) and (2) have been implemented in a limited range of research studies on the quantum fractional modelings. This yields the novelty and our main motivation to finalize this manuscript.

This research scheme is outlined as follows: We present the main concepts of the quantum calculus in Section 2. Our main results caused by new fixed-point approaches about solutions' existence of quantum BVP (1) and (2) will be obtained in Section 3. In Section 4, two numerical examples will be provided to support and validate our obtained results. A conclusion about our research work will be stated in Section 5.

## 2. Fundamental Preliminaries

In this section, some important issues in the sense of q-calculus are discussed. We suppose that $0 < q < 1$. On the function $(m_1 - m_2)^n$ given for $n \in \mathbb{N}_0$, its q-analogue is defined by $(m_1 - m_2)^{(0)} = 1$, and

$$(m_1 - m_2)^{(n)} = \prod_{k=0}^{n-1} (m_1 - m_2 q^k),$$

so that $m_1, m_2 \in \mathbb{R}$ and $\mathbb{N}_0 := \{0, 1, 2, \dots\}$ [17]. Now, $n = \varsigma$ is a constant which is assumed to be contained in $\mathbb{R}$. Let us now display the follwoing q-analogue of the existing power mapping $(m_1 - m_2)^n$ in a q-fractional settings:

$$(m_1 - m_2)^{(\varsigma)} = m_1^\varsigma \prod_{n=0}^{\infty} \frac{1 - (\frac{m_2}{m_1})q^n}{1 - (\frac{m_2}{m_1})q^{\varsigma+n}}, \tag{3}$$

for $m_1 \neq 0$. We note that by having $m_2 = 0$, an equality $m_1^{(\varsigma)} = m_1^\varsigma$ is obtained immediately [17]. For the given real number $m_1 \in \mathbb{R}$, a q-number $[m_1]_q$ is expressed as:

$$[m_1]_q = \frac{1 - q^{m_1}}{1 - q} = q^{m_1 - 1} + \cdots + q + 1.$$

The q-Gamma function is illustrated using the following format:

$$\Gamma_q(r) = \frac{(1-q)^{(r-1)}}{(1-q)^{r-1}}, \tag{4}$$

so that $r \in \mathbb{R} \setminus \{0, -1, -2, \dots\}$ [9,17]. It is notable that $\Gamma_q(r+1) = [r]_q \Gamma_q(r)$ is valid [9]. A pseudo-code inspired by (3) and (4) is proposed in Algorithm 1 for computing various Gamma function's values in the proposed quantum settings.

Given a real-valued continuous function $\hbar$, the quantum derivative of this function can be formulated by:

$$(\,_q\mathfrak{D}_{0^+}\hbar)(r) = \frac{\hbar(r) - \hbar(qr)}{(1-q)r}, \tag{5}$$

and also $(\,_q\mathfrak{D}_{0^+}\hbar)(0) = \lim_{r \to 0}(\,_q\mathfrak{D}_{0^+}\hbar)(r)$ [34]. Given a function $\hbar$, the quantum derivative of this function can be extended to an arbitrary higher order by $(\,_q\mathfrak{D}^n_{0^+}\hbar)(r) =$

$_q\mathfrak{D}_{0^+}(\ _q\mathfrak{D}_{0^+}^{n-1}\hbar)(r)$ for any $n \in \mathbb{N}$ [34]. Obviously, we notice that $(\ _q\mathfrak{D}_{0^+}^0\hbar)(r) = \hbar(r)$. Similarly, for computing this kind of q-derivative of $\hbar$, in Algorithm 2, we propose a pseudo-code inspired by (5).

---

**Algorithm 1** Pseudo-code for $\Gamma_q(\varsigma)$:

---

**Require:** $\varsigma \in \mathbb{R}\backslash\{0\} \cup \mathbb{Z}^-, q \in (0,1), n$
 1: $w \leftarrow 1$
 2: **for** $l = 0$ to $n$ **do**
 3:  $w \leftarrow w((1-q^{l+1})/(1-q^{\varsigma+l}))$
 4: **end for**
 5: $\Gamma_q(\varsigma) \leftarrow w/(1-q)^{\varsigma-1}$
**Ensure:** $\Gamma_q(\varsigma)$

---

**Algorithm 2** Pseudo-code for $_q\mathfrak{D}_{0^+}\hbar(r)$:

---

**Require:** $q \in (0,1), \hbar(r), r$
 1: syms $b$
 2: **if** $r = 0$ **then**
 3:  $\phi \leftarrow \lim((\hbar(b) - \hbar(q * b))/((1-q)b), b, 0)$
 4: **else**
 5:  $\phi \leftarrow (\hbar(r) - \hbar(q * r))/((1-q) * r)$
 6: **end if**
**Ensure:** $_q\mathfrak{D}_{0^+}\hbar(r)$

---

Given continuous map $\hbar : [0, m_2] \to \mathbb{R}$, the quantum integral of this function can be expressed as:

$$(\ _q\mathfrak{I}_{0^+}\hbar)(r) = \int_0^r \hbar(v)\,\mathrm{d}_q v = r(1-q)\sum_{k=0}^\infty \hbar(rq^k)q^k, \quad (r \in [0, m_2]) \tag{6}$$

provided the absolute convergence of the existing series holds [34]. The quantum integral of $\hbar$ can be similarly extended like quantum derivative to an arbitrary higher order using an iterative rule $(\ _q\mathfrak{I}_{0^+}^n\hbar)(r) = \ _q\mathfrak{I}_{0^+}(\ _q\mathfrak{I}_{0^+}^{n-1}\hbar)(r)$ for all $n \geq 1$ [34]. Moreover, it is clear to note that $(\ _q\mathfrak{I}_{0^+}^0\hbar)(r) = \hbar(r)$. A pseudo-code caused by (6) is proposed in in Algorithm 3. We now suppose that $m_1 \in [0, m_2]$. This time, the similar q-operator of $\hbar$ from $m_1$ to $m_2$ can be defined in this case as follows:

$$\int_{m_1}^{m_2} \hbar(v)\,\mathrm{d}_q v = \ _q\mathfrak{I}_{0^+}\hbar(m_2) - \ _q\mathfrak{I}_{0^+}\hbar(m_1)$$

$$= \int_0^{m_2} \hbar(v)\,\mathrm{d}_q v - \int_0^{m_1} \hbar(v)\,\mathrm{d}_q v$$

$$= (1-q)\sum_{k=0}^\infty [m_2\hbar(m_2 q^k) - m_1\hbar(m_1 q^k)]q^k, \tag{7}$$

when the series exists [34]. A proposed pseudo-code caused by (7) is organized in Algorithm 4 for such a purpose.

If we assume that a function $\hbar$ is continuous at $r = 0$, then $(\ _q\mathfrak{I}_{0^+}\ _q\mathfrak{D}_{0^+}\hbar)(r) = \hbar(r) - \hbar(0)$ is obtained [34]. Moreover, the equality $(\ _q\mathfrak{D}_{0^+}\ _q\mathfrak{I}_{0^+}\hbar)(r) = \hbar(r)$ holds for each $r$. By considering a real number $\varsigma \geq 0$ in this case such that $n - 1 < \varsigma < n$, i.e., $n = [\varsigma] + 1$, for given function $\hbar \in \mathcal{C}_\mathbb{R}([0, +\infty))$, the RLQI of $\hbar$ is introduced by:

$$_q^R\mathfrak{I}_{0^+}^\varsigma\hbar(r) = \frac{1}{\Gamma_q(\varsigma)}\int_0^r (r - qv)^{(\varsigma-1)}\hbar(v)\,\mathrm{d}_q v, \quad \varsigma > 0$$

provided that the above value is finite and $^R_q\mathfrak{J}^0_{0+}\hbar(r) = \hbar(r)$ [35,36]. Further, the semi-group specification for the mentioned q-operator occurs such that $^R_q\mathfrak{J}^{\varsigma_1}_{0+}\left(^R_q\mathfrak{J}^{\varsigma_2}_{0+}\hbar\right)(r) = {}^R_q\mathfrak{J}^{\varsigma_1+\varsigma_2}_{0+}\hbar(r)$ for $\sigma_1, \sigma_2 \geq 0$ [35]. For $\theta \in (-1, \infty)$,

$$^R_q\mathfrak{J}^{\varsigma}_{0+}r^\theta = \frac{\Gamma_q(\theta+1)}{\Gamma_q(\theta+\varsigma+1)}r^{\theta+\varsigma}, \quad (r > 0).$$

It is evident that if we take $\theta = 0$, then $^R_q\mathfrak{J}^{\varsigma}_{0+}1(r) = \dfrac{1}{\Gamma_q(\varsigma+1)}r^{\varsigma}$ for any $r > 0$. Given a function $\hbar \in \mathcal{C}^{(n)}_{\mathbb{R}}([0, +\infty))$, the CpQD for this function is formulated by:

$$^C_q\mathfrak{D}^{\varsigma}_{0+}\hbar(r) = \frac{1}{\Gamma_q(n-\varsigma)}\int_0^r (r-qv)^{(n-\varsigma-1)}\,{}_q\mathfrak{D}^n_{0+}\hbar(v)\,\mathrm{d}_qv,$$

if the integral exists [35,36]. The following property is valid:

$$^C_q\mathfrak{D}^{\varsigma}_{0+}r^\theta = \frac{\Gamma_q(\theta+1)}{\Gamma_q(\theta-\varsigma+1)}r^{\theta-\varsigma}, \quad (r > 0).$$

It is evident that $^C_q\mathfrak{D}^{\varsigma}_{0+}1(r) = 0$ for any $r > 0$. For instance, by letting $\theta = 2$, $q = 0.5$ and $\hbar(r) = r^2$, we have

$$^C_{0.5}\mathfrak{D}^{\varsigma}_{0+}r^2 = \frac{\Gamma_{0.5}(3)}{\Gamma_{0.5}(3-\varsigma)}r^{2-\varsigma}.$$

In this direction, the graph of the CpQD for the function $\hbar(r) = r^2$ for $q = 0.5$ is available in Figure 1.

---

**Algorithm 3** Pseudo-code for $_q\mathfrak{J}^{\varsigma}_{0+}\hbar(r)$:

---

**Require:** $\varsigma, n, \hbar(r), r, q \in (0,1)$
  1: $P \leftarrow 0$
  2: **for** $k = 0$ to $n$ **do**
  3:   $\phi \leftarrow (1-q^{k+1})^{\varsigma-1}$
  4:   $P \leftarrow P + \phi * q^k * \hbar(r * q^k)$
  5: **end for**
  6: $\psi \leftarrow (r^{\varsigma} * (1-q) * P)/(\Gamma_q(r))$
**Ensure:** $_q\mathfrak{J}^{\varsigma}_{0+}\hbar(r)$

---

---

**Algorithm 4** Pseudo-code for $\displaystyle\int_{m_1}^{m_2}\hbar(v)\,\mathrm{d}_qv$:

---

**Require:** $\hbar(r), m_1, k, m_2, q \in (0,1)$
  1: $P \leftarrow 0$
  2: **for** $l = 0 : k$ **do**
  3:   $P \leftarrow P + q^l * (m_2 * \hbar(m_2 * q^l) - m_1 * \hbar(m_1 * q^l))$
  4: **end for**
  5: $\phi \leftarrow (1-q) * P$
**Ensure:** $\displaystyle\int_{m_1}^{m_2}\hbar(v)\,\mathrm{d}_qv$

---

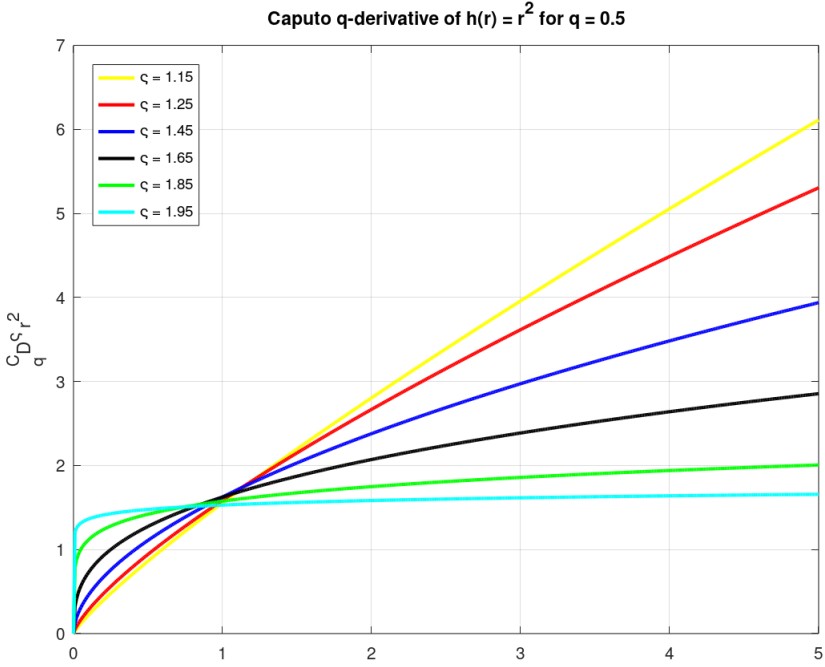

**Figure 1.** The graph of the Caputo q-derivative of $\hbar(r) = r^2$ for $q = 0.5$.

**Lemma 1** ([37]). *Assume that $n - 1 < \varsigma < n$ and $\hbar \in \mathcal{C}_{\mathbb{R}}^{(n)}([0, +\infty))$. Then, we have:*

$$\left( {}_q^C\mathfrak{I}_{0^+}^\varsigma \, {}_q^C\mathfrak{D}_{0^+}^\varsigma \hbar \right)(r) = \hbar(r) - \sum_{k=0}^{n-1} \frac{r^k}{\Gamma_q(k+1)} \left( {}_q\mathfrak{D}_{0^+}^k \hbar \right)(0).$$

According to the above lemma, the given fractional quantum differential equation, ${}_q^C\mathfrak{D}_{0^+}^\varsigma \hbar(r) = 0$, has a general solution which is obtained by $\hbar(r) = \tilde{\mu}_0 + \tilde{\mu}_1 r + \tilde{\mu}_2 r^2 + \cdots + \tilde{\mu}_{n-1} r^{n-1}$ so that $\tilde{\mu}_0, \ldots, \tilde{\mu}_{n-1} \in \mathbb{R}$, and $n = [\varsigma] + 1$ [37]. It is worth noting that for each continuous $\hbar$, according to Lemma 1, we get:

$$\left( {}_q^R\mathfrak{I}_{0^+}^\varsigma \, {}_q^C\mathfrak{D}_{0^+}^\varsigma \hbar \right)(r) = \hbar(r) + \tilde{\mu}_0 + \tilde{\mu}_1 r + \tilde{\mu}_2 r^2 + \cdots + \tilde{\mu}_{n-1} r^{n-1},$$

where $\tilde{\mu}_0, \ldots, \tilde{\mu}_{n-1}$ illustrate constants contained in $\mathbb{R}$, and $n = [\varsigma] + 1$ [37].

Next, we recall some essential inequalities and concepts. The Kuratowski measure of noncompactness $\mathbb{O}$ is defined by

$$\mathbb{O}(\mathcal{H}) := \inf\left\{ \varepsilon > 0 : \mathcal{H} = \bigcup_{k=1}^{n} \mathcal{H}_k \text{ and diam } (\mathcal{H}_k) \leq \varepsilon \text{ for } k = 1, \ldots, n \right\},$$

where $\text{diam}(\mathcal{H}_k) = \sup\{|\hbar - \hbar'| : \hbar, \hbar' \in \mathcal{H}_k\}$ and $\mathcal{H}$ is bounded subset of Banach space $\mathfrak{A}$. Moreover, it is identified that $0 \leq \mathbb{O}(\mathcal{H}) \leq \text{diam}(\mathcal{H}) < +\infty$ [38].

**Lemma 2** ([38]). *Consider the bounded subsets $\mathcal{H}, \mathcal{H}_1$ and $\mathcal{H}_2$ of an arbitrary real Banach space $\mathfrak{A}$. Then, the following conditions hold:*

$(\mathbb{C}_1)$ $\mathbb{O}(\mathcal{H}) = 0$ *iff $\mathcal{H}$ is precompact;*

$(\mathbb{C}_2)$ $\mathbb{O}(\mathcal{H}) = \mathbb{O}(\bar{\mathcal{H}}) = \mathbb{O}(\text{cnvx}(\mathcal{H}))$, *where $\bar{\mathcal{H}}$ and $\text{cnvx}(\mathcal{H})$ are the closure and convex hull of $\mathcal{H}$;*

$(\mathbb{C}_3)$ *if $\mathcal{H}_1 \subseteq \mathcal{H}_2$, then $\mathbb{O}(\mathcal{H}_1) \leq \mathbb{O}(\mathcal{H}_2)$;*

$(\mathbb{C}_4)$ $\forall \kappa \in \mathbb{R}$, $\mathbb{O}(\kappa + \mathcal{H}) \leq \mathbb{O}(\mathcal{H})$;

$(\mathbb{C}_5)$ $\forall \kappa \in \mathbb{R}$, $\mathbb{O}(\kappa\mathcal{H}) = |\kappa|\mathbb{O}(\mathcal{H})$;

$(\mathbb{C}_6)$ $\mathbb{O}(\mathcal{H}_1 + \mathcal{H}_2) \leq \mathbb{O}(\mathcal{H}_1) + \mathbb{O}(\mathcal{H}_2)$, where $\mathcal{H}_1 + \mathcal{H}_2 = \{\hbar_1 + \hbar_2; \hbar_1 \in \mathcal{H}_1, \hbar_2 \in \mathcal{H}_2\}$;

$(\mathbb{C}_7)$ $\mathbb{O}(\mathcal{H}_1 \cup \mathcal{H}_2) \leq \max\{\mathbb{O}(\mathcal{H}_1) + \mathbb{O}(\mathcal{H}_2)\}$.

**Lemma 3** ([39])**.** *Regard $\mathfrak{A}$ as a Banach space. Then, for each bounded set $\mathcal{H} \subseteq \mathfrak{A}$, a countable set $\mathcal{H}_0 \subseteq \mathcal{H}$ exists subject to $\mathbb{O}(\mathcal{H}) \leq 2\,\mathbb{O}(\mathcal{H}_0)$.*

**Lemma 4** ([38])**.** *Regard $\mathfrak{A}$ as a Banach space. Let $\mathcal{H}$ be bounded and equi-continuous set contained in $\mathcal{C}_{\mathfrak{A}}([a,b])$. Then, $\mathbb{O}(\mathcal{H}(r))$ is continuous on $[a,b]$, and we have $\mathbb{O}(\mathcal{H}) = \sup_{r\in[a,b]} \mathbb{O}(\mathcal{H}(r))$.*

**Lemma 5** ([38])**.** *Let $\mathfrak{A}$ be a Banach space. Let $\mathcal{H} = \{\hbar_n\}_{n\geq 1} \subseteq \mathcal{C}_{\mathfrak{A}}([a,b])$ be bounded and countable set. Then, $\mathbb{O}(\mathcal{H}(r))$ is Lebesgue integrable on $[a,b]$, and we have:*

$$\mathbb{O}\left(\left\{\int_0^r \hbar_n(v)\,dv\right\}_{n\geq 1}\right) \leq 2\int_0^r \mathbb{O}(\{\hbar_n(v)\}_{n\geq 1})\,dv.$$

**Definition 1** ([38])**.** *Regard $\mathfrak{A}$ as a Banach space and $\varphi_* : \mathcal{S} \subset \mathfrak{A} \to \mathfrak{A}$ as a bounded and continuous operator. Then, the map $\varphi_*$ is termed condensing if for any bounded closed set $\mathcal{H} \subseteq \mathcal{S}$, the inequality $\mathbb{O}(\varphi_*(\mathcal{H})) < \mathbb{O}(\mathcal{H})$ holds.*

**Theorem 1** ([38], Sadovskii's fixed point theorem)**.** *Regard $\mathfrak{A}$ as a Banach space. Let $\mathcal{H}$ be a bounded, closed and convex set contained in $\mathfrak{A}$. Furthermore, assume that continuous mapping $\varphi_* : \mathcal{H} \to \mathcal{H}$ is condensing. Then, there exists at least one fixed point for the map $\varphi_*$ in $\mathcal{H}$.*

Let us denote the normed space by $(\mathfrak{A}, \|\cdot\|_{\mathfrak{A}})$. Regard $\mathbb{P}(\mathfrak{A}), \mathbb{P}_{bd}(\mathfrak{A}), \mathbb{P}_{cl}(\mathfrak{A}), \mathbb{P}_{cm}(\mathfrak{A})$ and $\mathbb{P}_{cx}(\mathfrak{A})$ as a family of all non-empty, all bounded, all closed, all compact and all convex sets contained in $\mathfrak{A}$, respectively.

**Definition 2** ([40])**.** *An element $\hbar \in \mathfrak{A}$ is termed an endpoint of a multi-valued function $\mathbb{T}_* : \mathfrak{A} \to \mathbb{P}(\mathfrak{A})$ whenever we get $\mathbb{T}_*(\hbar) = \{\hbar\}$.*

The multi-valued map $\mathbb{T}_*$ has an approximate endpoint criterion (AEPC) if

$$\inf_{\hbar_1 \in \mathfrak{A}} \sup_{\hbar_2 \in \mathbb{T}_*(\hbar_1)} d(\hbar_1, \hbar_2) = 0,$$

Ref. [40]. Next, a required theorem related to the proposed quantum boundary problem is recalled.

**Theorem 2** ([40], Endpoint theorem)**.** *Let's assume that $(\mathfrak{A}, d)$ is a complete metric space, and $\psi : [0,\infty) \to [0,\infty)$ is u.s.c subject to for each $r > 0$, $\liminf_{r\to\infty}(r - \psi(r)) > 0$, and $\psi(r) < r$. Assume that $\mathbb{T}_* : \mathfrak{A} \to \mathbb{P}_{cl,bd}(\mathfrak{A})$ is a multi-valued map such that for each $\hbar_1, \hbar_2 \in \mathfrak{A}$, the following inequality holds:*
$$\mathbb{H}_d(\mathbb{T}_*\hbar_1, \mathbb{T}_*\hbar_2) \leq \psi(d(\hbar_1, \hbar_2)).$$

*Then, there is exactly one endpoint for $\mathbb{T}_*$ iff $\mathbb{T}_*$ has an approximate endpoint criterion.*

## 3. Main Results

We regard the family of continuous functions on $[0,1]$ by $\mathfrak{A} = \mathcal{C}_{\mathbb{R}}([0,1])$ and the defined sup-norm $\|\hbar\|_{\mathfrak{A}} = \sup_{r\in[0,1]} |\hbar(r)|$, for all members $\hbar \in \mathfrak{A}$, confirms that the space $\mathfrak{A}$ becomes a Banach space. In the sequel, we will establish the existence results for quantum BVP (1) and (2). Before moving to the existence results, the following proposition will play an essential role:

**Proposition 1.** *Let* $\varphi_* \in \mathfrak{A}$, $\varsigma \in (2,3)$, $\varrho \in (1,2)$, $\xi \in (0,1)$, $\ell_1, \ell_2, \ell_3 \in \mathbb{R}^{>0}$ *and* $\sigma > 0$. *Then, the function* $\hbar^*$ *satisfies as a solution for the given quantum integro-difference FBVP (CpQFP) formulated by*

$$
\begin{cases}
{}^C_q\mathfrak{D}^\varsigma_{0^+}\hbar^*(r) = \varphi_*(r), & (r \in [0,1],\ q \in (0,1)), \\[2mm]
\hbar(0) + \hbar(\xi) = \ell_1 \,{}^R_q\mathfrak{I}^\sigma_{0^+}\hbar(1), \\[2mm]
{}^C_q\mathfrak{D}^\varrho_{0^+}\hbar(0) + {}^C_q\mathfrak{D}^\varrho_{0^+}\hbar(\xi) = \ell_2 \,{}^R_q\mathfrak{I}^\sigma_{0^+}\left[{}^C_q\mathfrak{D}^\varrho_{0^+}\hbar\right](1), \\[2mm]
{}^C_q\mathfrak{D}^1_{0^+}\hbar(0) + {}^C_q\mathfrak{D}^1_{0^+}\hbar(\xi) = \ell_3 \,{}^R_q\mathfrak{I}^\sigma_{0^+}\left[{}^C_q\mathfrak{D}^1_{0^+}\hbar\right](1),
\end{cases}
\tag{8}
$$

*iff* $\hbar^*$ *is a solution for the fractional quantum integral (FQI) equation given by*

$$
\hbar^*(r) = \int_0^r \frac{(r-qv)^{(\varsigma-1)}}{\Gamma_q(\varsigma)}\varphi_*(v)\,\mathrm{d}_q v + \frac{\ell_1}{\delta_1}\int_0^1 \frac{(1-qv)^{(\varsigma+\sigma-1)}}{\Gamma_q(\varsigma+\sigma)}\varphi_*(v)\,\mathrm{d}_q v
\tag{9}
$$

$$
- \frac{1}{\delta_1}\int_0^\xi \frac{(\xi-qv)^{(\varsigma-1)}}{\Gamma_q(\varsigma)}\varphi_*(v)\,\mathrm{d}_q v + \ell_3\Lambda_1(r)\int_0^1 \frac{(1-qv)^{(\varsigma+\sigma-2)}}{\Gamma_q(\varsigma+\sigma-1)}\varphi_*(v)\,\mathrm{d}_q v
$$

$$
- \Lambda_1(r)\int_0^\xi \frac{(\xi-qv)^{(\varsigma-2)}}{\Gamma_q(\varsigma-1)}\varphi_*(v)\,\mathrm{d}_q v
$$

$$
+ \ell_2\Lambda_2(r)\int_0^1 \frac{(1-qv)^{(\varsigma+\sigma-\varrho-1)}}{\Gamma_q(\varsigma+\sigma-\varrho)}\varphi_*(v)\,\mathrm{d}_q v - \Lambda_2(r)\int_0^\xi \frac{(\xi-qv)^{(\varsigma-\varrho-1)}}{\Gamma_q(\varsigma-\varrho)}\varphi_*(v)\,\mathrm{d}_q v.
\tag{10}
$$

**Proof.** Firstly, the given function $\hbar^*$ is regarded as a solution for (8). By virtue of $\varsigma \in (2,3)$, taking the integral in the RL-settings of order $\varsigma$ to (8), we arrive at

$$
\hbar^*(r) = \int_0^r \frac{(r-qv)^{(\varsigma-1)}}{\Gamma_q(\varsigma)}\varphi_*(v)\,\mathrm{d}_q v + \tilde{\mu}_0 + \tilde{\mu}_1 r + \tilde{\mu}_2 r^2,
\tag{11}
$$

so that $\tilde{\mu}_0, \tilde{\mu}_1, \tilde{\mu}_2 \in \mathbb{R}$ are some constants that are needed to be obtained. By considering $\varrho \in (1,2)$, the following immediate results are obtained

$$
{}^C_q\mathfrak{D}^1_{0^+}\hbar^*(r) = \int_0^r \frac{(r-qv)^{(\varsigma-2)}}{\Gamma_q(\varsigma-1)}\varphi_*(v)\,\mathrm{d}_q v + \tilde{\mu}_1 + \tilde{\mu}_2(1+q)r,
$$

$$
{}^C_q\mathfrak{D}^\varrho_{0^+}\hbar^*(r) = \int_0^r \frac{(r-qv)^{(\varsigma-\varrho-1)}}{\Gamma_q(\varsigma-\varrho)}\varphi_*(v)\,\mathrm{d}_q v + \tilde{\mu}_2\frac{2r^{2-\varrho}}{\Gamma_q(3-\varrho)},
$$

$$
{}^R_q\mathfrak{I}^\sigma_{0^+}\hbar^*(r) = \int_0^r \frac{(r-qv)^{(\varsigma+\sigma-1)}}{\Gamma_q(\varsigma+\sigma)}\varphi_*(v)\,\mathrm{d}_q v + \tilde{\mu}_0\frac{r^\sigma}{\Gamma_q(\sigma+1)} + \tilde{\mu}_1\frac{r^{\sigma+1}}{\Gamma_q(\sigma+2)}
$$

$$
+ \tilde{\mu}_2\frac{(1+q)r^{\sigma+2}}{\Gamma_q(\sigma+3)},
$$

$$
{}^R_q\mathfrak{I}^\sigma_{0^+}\left({}^C_q\mathfrak{D}^1_{0^+}\hbar^*(r)\right) = \int_0^r \frac{(r-qv)^{(\varsigma+\sigma-2)}}{\Gamma_q(\varsigma+\sigma-1)}\varphi_*(v)\,\mathrm{d}_q v + \tilde{\mu}_1\frac{r^\sigma}{\Gamma_q(\sigma+1)} + \tilde{\mu}_2\frac{(1+q)r^{\sigma+1}}{\Gamma_q(\sigma+2)},
$$

$$
{}^R_q\mathfrak{I}^\sigma_{0^+}\left({}^C_q\mathfrak{D}^\varrho_{0^+}\hbar^*(r)\right) = \int_0^r \frac{(r-qv)^{(\varsigma+\sigma-\varrho-1)}}{\Gamma_q(\varsigma+\sigma-\varrho)}\varphi_*(v)\,\mathrm{d}_q v + \tilde{\mu}_2\frac{2r^{\sigma+2-\varrho}}{\Gamma_q(\sigma+3-\varrho)}.
$$

Now, by virtue of the given boundary conditions, we get

$$
\tilde{\mu}_0 = \frac{\ell_1}{\delta_1}\int_0^1 \frac{(1-qv)^{(\varsigma+\sigma-1)}}{\Gamma_q(\varsigma+\sigma)}\varphi_*(v)\,\mathrm{d}_q v - \frac{1}{\delta_1}\int_0^\xi \frac{(\xi-qv)^{(\varsigma-1)}}{\Gamma_q(\varsigma)}\varphi_*(v)\,\mathrm{d}_q v
$$

$$- \ell_3 \Theta_1 \int_0^1 \frac{(1 - qv)^{(\varsigma + \sigma - 2)}}{\Gamma_q(\varsigma + \sigma - 1)} \varphi_*(v) \, \mathrm{d}_q v + \Theta_1 \int_0^\xi \frac{(\xi - qv)^{(\varsigma - 2)}}{\Gamma_q(\varsigma - 1)} \varphi_*(v) \, \mathrm{d}_q v$$

$$+ \ell_2 \Theta_2 \int_0^1 \frac{(1 - qv)^{(\varsigma + \sigma - \varrho - 1)}}{\Gamma_q(\varsigma + \sigma - \varrho)} \varphi_*(v) \, \mathrm{d}_q v - \Theta_2 \int_0^\xi \frac{(\xi - qv)^{(\varsigma - \varrho - 1)}}{\Gamma_q(\varsigma - \varrho)} \varphi_*(v) \, \mathrm{d}_q v,$$

$$\tilde{\mu}_1 = \frac{\ell_3}{\Delta_1} \int_0^1 \frac{(1 - qv)^{(\varsigma + \sigma - 2)}}{\Gamma_q(\varsigma + \sigma - 1)} \varphi_*(v) \, \mathrm{d}_q v - \frac{1}{\Delta_1} \int_0^\xi \frac{(\xi - qv)^{(\varsigma - 2)}}{\Gamma_q(\varsigma - 1)} \varphi_*(v) \, \mathrm{d}_q v$$

$$- \frac{\ell_2 \Delta_2}{\Delta_1 \Delta_3} \int_0^1 \frac{(1 - qv)^{(\varsigma + \sigma - \varrho - 1)}}{\Gamma_q(\varsigma + \sigma - \varrho)} \varphi_*(v) \, \mathrm{d}_q v + \frac{\Delta_2}{\Delta_1 \Delta_3} \int_0^\xi \frac{(\xi - qv)^{(\varsigma - \varrho - 1)}}{\Gamma_q(\varsigma - \varrho)} \varphi_*(v) \, \mathrm{d}_q v,$$

and

$$\tilde{\mu}_2 = \frac{\ell_2}{\Delta_3} \int_0^1 \frac{(1 - qv)^{(\varsigma + \sigma - \varrho - 1)}}{\Gamma_q(\varsigma + \sigma - \varrho)} \varphi_*(v) \, \mathrm{d}_q v - \frac{1}{\Delta_3} \int_0^\xi \frac{(\xi - qv)^{(\varsigma - \varrho - 1)}}{\Gamma_q(\varsigma - \varrho)} \varphi_*(v) \, \mathrm{d}_q v,$$

where we regard the constants

$$\delta_1 = \frac{2\Gamma_q(\sigma + 1) - \ell_1}{\Gamma_q(\sigma + 1)}, \quad \delta_2 = \frac{\xi \Gamma_q(\sigma + 2) - \ell_1}{\Gamma_q(\sigma + 2)}, \quad \delta_3 = \frac{\xi^2 \Gamma_q(\sigma + 3) - \ell_1(1 + q)}{\Gamma_q(\sigma + 3)},$$

$$\Delta_1 = \frac{2\Gamma_q(\sigma + 1) - \ell_3}{\Gamma_q(\sigma + 1)}, \quad \Delta_2 = \frac{(1 + q)(\xi \Gamma_q(\sigma + 2) - \ell_3)}{\Gamma_q(\sigma + 2)},$$

$$\Delta_3 = \frac{2\xi^{2 - \varrho} \Gamma_q(\sigma + 3 - \varrho) - 2\ell_2 \Gamma_q(3 - \varrho)}{\Gamma_q(3 - \varrho) \Gamma_q(\sigma + 3 - \varrho)}, \quad \Theta_1 = \frac{\delta_2}{\delta_1 \Delta_1}, \quad \Theta_2 = \frac{\delta_2 \Delta_2 - \delta_3 \Delta_1}{\delta_1 \Delta_1 \Delta_3},$$

along with the functions with respect to $r$ as

$$\Lambda_1(r) = \frac{r - \Theta_1 \Delta_1}{\Delta_1}, \qquad \Lambda_2(r) = \frac{r^2 \Delta_1 - r \Delta_2 + \Theta_2 \Delta_1 \Delta_3}{\Delta_1 \Delta_3}. \tag{12}$$

By substituting the values of $\tilde{\mu}_0$, $\tilde{\mu}_1$ and $\tilde{\mu}_2$ in (11), integral solution (9) is obtained. The converse part can be easily deduced. □

**Remark 1.** *Note that for simplicity in the subsequent computations, we set the following upper bounds by virtue of the functions displayed in* (12):

$$|\Lambda_1(r)| \leq \frac{1 + |\Theta_1||\Delta_1|}{|\Delta_1|} := \Lambda_1^* > 0,$$

$$|\Lambda_2(r)| \leq \frac{|\Delta_1| + |\Delta_2| + |\Theta_2||\Delta_1||\Delta_3|}{|\Delta_1||\Delta_3|} := \Lambda_2^* > 0. \tag{13}$$

**Theorem 3.** *Let* $\varphi_* : [0, 1] \times \mathfrak{A} \to \mathbb{R}$ *be continuous. In addition, assume that there exists a continuous* $\vartheta : [0, 1] \to \mathbb{R}^{>0}$ *along with a nondecreasing continuous map* $\wp : [0, \infty) \to (0, \infty)$ *such that for each* $r \in [0, 1]$ *and* $\hbar \in \mathfrak{A}$,

$$|\varphi_*(r, \hbar(r))| \leq \vartheta(r) \wp(\|\hbar\|_{\mathfrak{A}}). \tag{14}$$

*We suppose that there exists a function* $m_{\varphi_*} : [0, 1] \to \mathbb{R}$ *such that for each bounded set* $\mathcal{H} \subseteq \mathfrak{A}$ *and* $r \in [0, 1]$,

$$\mathbb{O}(\varphi_*(r, \mathcal{H})) \leq m_{\varphi_*}(r) \mathbb{O}(\mathcal{H}). \tag{15}$$

*Then, at least one solution of the given Caputo fractional quantum BVP* (1) *exists on* $[0, 1]$ *if*

$$\left[ \frac{\tilde{m}_{\varphi_*}}{\Gamma_q(\varsigma + 1)} + \frac{\tilde{m}_{\varphi_*}}{|\delta_1|} \left( \frac{\ell_1}{\Gamma_q(\varsigma + \sigma + 1)} + \frac{\xi^{(\varsigma)}}{\Gamma_q(\varsigma + 1)} \right) + \tilde{m}_{\varphi_*} \Lambda_1^* \left( \frac{\ell_3}{\Gamma_q(\varsigma + \sigma)} + \frac{\xi^{(\varsigma - 1)}}{\Gamma_q(\varsigma)} \right) \right.$$

$$\left. + \tilde{m}_{\varphi_*} \Lambda_2^* \left( \frac{\ell_2}{\Gamma_q(\varsigma + \sigma - \varrho + 1)} + \frac{\xi^{(\varsigma - \varrho)}}{\Gamma_q(\varsigma - \varrho + 1)} \right) \right] < \frac{1}{4}, \tag{16}$$

*where* $\tilde{m}_{\varphi_*} = \sup_{r \in [0,1]} |m_{\varphi_*}(r)|$.

**Proof.** Introduce the mapping $\mathcal{G} : \mathfrak{H} \to \mathfrak{H}$ defined as:

$$\mathcal{G}(\hbar)(r) = \int_0^r \frac{(r - qv)^{(\varsigma - 1)}}{\Gamma_q(\varsigma)} \varphi_*(v, \hbar(v)) \, \mathrm{d}_q v \tag{17}$$

$$+ \frac{\ell_1}{\delta_1} \int_0^1 \frac{(1 - qv)^{(\varsigma + \sigma - 1)}}{\Gamma_q(\varsigma + \sigma)} \varphi_*(v, \hbar(v)) \, \mathrm{d}_q v - \frac{1}{\delta_1} \int_0^\xi \frac{(\xi - qv)^{(\varsigma - 1)}}{\Gamma_q(\varsigma)} \varphi_*(v, \hbar(v)) \, \mathrm{d}_q v$$

$$+ \ell_3 \Lambda_1(r) \int_0^1 \frac{(1 - qv)^{(\varsigma + \sigma - 2)}}{\Gamma_q(\varsigma + \sigma - 1)} \varphi_*(v, \hbar(v)) \, \mathrm{d}_q v \tag{18}$$

$$- \Lambda_1(r) \int_0^\xi \frac{(\xi - qv)^{(\varsigma - 2)}}{\Gamma_q(\varsigma - 1)} \varphi_*(v, \hbar(v)) \, \mathrm{d}_q v$$

$$+ \ell_2 \Lambda_2(r) \int_0^1 \frac{(1 - qv)^{(\varsigma + \sigma - \varrho - 1)}}{\Gamma_q(\varsigma + \sigma - \varrho)} \varphi_*(v, \hbar(v)) \, \mathrm{d}_q v \tag{19}$$

$$- \Lambda_2(r) \int_0^\xi \frac{(\xi - qv)^{(\varsigma - \varrho - 1)}}{\Gamma_q(\varsigma - \varrho)} \varphi_*(v, \hbar(v)) \, \mathrm{d}_q v,$$

where $\mathfrak{H} = \{ \hbar \in \mathfrak{A} : \|\hbar\|_{\mathfrak{A}} \leq \varepsilon_*, \varepsilon_* \in \mathbb{R}^{>0} \} \subseteq \mathfrak{A}$ and is classified as a convex bounded closed space. Obviously, the fixed point of the proposed operator $\mathcal{G}$ is the quantum fractional BVP's solution (1).

Firstly, we verify the continuity of $\mathcal{G}$ on $\mathfrak{H}$. Take the sequence $\{\hbar_n\}_{n \geq 1}$ in $\mathfrak{H}$ such that $\hbar_n \to \hbar$ for each $\hbar \in \mathfrak{H}$. Since $\varphi_*$ is continuous on $[0, 1] \times \mathfrak{A}$, so we can write $\lim_{n \to \infty} \varphi_*(r, \hbar_n(r)) = \varphi_*(r, \hbar(r))$. Now, with the aid of Lebesgue dominated convergence theorem, we obtain:

$$\lim_{n \to \infty} (\mathcal{G} \hbar_n)(r) = \int_0^r \frac{(r - qv)^{(\varsigma - 1)}}{\Gamma_q(\varsigma)} \lim_{n \to \infty} \varphi_*(v, \hbar_n(v)) \, \mathrm{d}_q v$$

$$+ \frac{\ell_1}{\delta_1} \int_0^1 \frac{(1 - qv)^{(\varsigma + \sigma - 1)}}{\Gamma_q(\varsigma + \sigma)} \lim_{n \to \infty} \varphi_*(v, \hbar_n(v)) \, \mathrm{d}_q v$$

$$- \frac{1}{\delta_1} \int_0^\xi \frac{(\xi - qv)^{(\varsigma - 1)}}{\Gamma_q(\varsigma)} \lim_{n \to \infty} \varphi_*(v, \hbar_n(v)) \, \mathrm{d}_q v$$

$$+ \ell_3 \Lambda_1(r) \int_0^1 \frac{(1 - qv)^{(\varsigma + \sigma - 2)}}{\Gamma_q(\varsigma + \sigma - 1)} \lim_{n \to \infty} \varphi_*(v, \hbar_n(v)) \, \mathrm{d}_q v$$

$$- \Lambda_1(r) \int_0^\xi \frac{(\xi - qv)^{(\varsigma - 2)}}{\Gamma_q(\varsigma - 1)} \lim_{n \to \infty} \varphi_*(v, \hbar_n(v)) \, \mathrm{d}_q v$$

$$+ \ell_2 \Lambda_2(r) \int_0^1 \frac{(1 - qv)^{(\varsigma + \sigma - \varrho - 1)}}{\Gamma_q(\varsigma + \sigma - \varrho)} \lim_{n \to \infty} \varphi_*(v, \hbar_n(v)) \, \mathrm{d}_q v$$

$$- \Lambda_2(r) \int_0^{\xi} \frac{(\xi - qv)^{(\varsigma - \varrho - 1)}}{\Gamma_q(\varsigma - \varrho)} \lim_{n \to \infty} \varphi_*(v, \hbar_n(v)) \, \mathrm{d}_q v$$

$$= (\mathcal{G}\hbar)(r),$$

for each $r \in [0, 1]$. Thus, we get $\lim_{n \to \infty} (\mathcal{G}\hbar_n)(r) = (\mathcal{G}\hbar)(r)$. Hence, the continuity of $\mathcal{G}$ on $\mathfrak{H}$ is proved. Now, we want to examine uniform boundedness of $\mathcal{G}$ on $\mathfrak{H}$. To accomplish this goal, consider $\hbar \in \mathfrak{H}$. In view of inequalities (13) and (14), we have:

$$|(\mathcal{G}\hbar)(r)| = \int_0^r \frac{(r - qv)^{(\varsigma - 1)}}{\Gamma_q(\varsigma)} |\varphi_*(v, \hbar(v))| \, \mathrm{d}_q v$$

$$+ \frac{\ell_1}{|\delta_1|} \int_0^1 \frac{(1 - qv)^{(\varsigma + \sigma - 1)}}{\Gamma_q(\varsigma + \sigma)} |\varphi_*(v, \hbar(v))| \, \mathrm{d}_q v$$

$$+ \frac{1}{|\delta_1|} \int_0^{\xi} \frac{(\xi - qv)^{(\varsigma - 1)}}{\Gamma_q(\varsigma)} |\varphi_*(v, \hbar(v))| \, \mathrm{d}_q v$$

$$+ \ell_3 |\Lambda_1(r)| \int_0^1 \frac{(1 - qv)^{(\varsigma + \sigma - 2)}}{\Gamma_q(\varsigma + \sigma - 1)} |\varphi_*(v, \hbar(v))| \, \mathrm{d}_q v$$

$$+ |\Lambda_1(r)| \int_0^{\xi} \frac{(\xi - qv)^{(\varsigma - 2)}}{\Gamma_q(\varsigma - 1)} |\varphi_*(v, \hbar(v))| \, \mathrm{d}_q v$$

$$+ \ell_2 |\Lambda_2(r)| \int_0^1 \frac{(1 - qv)^{(\varsigma + \sigma - \varrho - 1)}}{\Gamma_q(\varsigma + \sigma - \varrho)} |\varphi_*(v, \hbar(v))| \, \mathrm{d}_q v$$

$$+ |\Lambda_2(r)| \int_0^{\xi} \frac{(\xi - qv)^{(\varsigma - \varrho - 1)}}{\Gamma_q(\varsigma - \varrho)} |\varphi_*(v, \hbar(v))| \, \mathrm{d}_q v$$

$$\leq \frac{1}{\Gamma_q(\varsigma + 1)} \vartheta(r) \wp(\|\hbar\|_{\mathfrak{A}}) + \frac{\ell_1}{|\delta_1| \Gamma_q(\varsigma + \sigma + 1)} \vartheta(r) \wp(\|\hbar\|_{\mathfrak{A}})$$

$$+ \frac{\xi^{(\varsigma)}}{|\delta_1| \Gamma_q(\varsigma + 1)} \vartheta(r) \wp(\|\hbar\|_{\mathfrak{A}})$$

$$+ \frac{\ell_3 \Lambda_1^*}{\Gamma_q(\varsigma + \sigma)} \vartheta(r) \wp(\|\hbar\|_{\mathfrak{A}}) + \frac{\Lambda_1^* \xi^{(\varsigma - 1)}}{\Gamma_q(\varsigma)} \vartheta(r) \wp(\|\hbar\|_{\mathfrak{A}})$$

$$+ \frac{\ell_2 \Lambda_2^*}{\Gamma_q(\varsigma + \sigma - \varrho + 1)} \vartheta(r) \wp(\|\hbar\|_{\mathfrak{A}}) + \frac{\Lambda_2^* \xi^{(\varsigma - \varrho)}}{\Gamma_q(\varsigma - \varrho + 1)} \vartheta(r) \wp(\|\hbar\|_{\mathfrak{A}}).$$

Set

$$\hat{\Omega} = \frac{1}{\Gamma_q(\varsigma + 1)} + \frac{1}{|\delta_1|} \left( \frac{\ell_1}{\Gamma_q(\varsigma + \sigma + 1)} + \frac{\xi^{(\varsigma)}}{\Gamma_q(\varsigma + 1)} \right) + \Lambda_1^* \left( \frac{\ell_3}{\Gamma_q(\varsigma + \sigma)} + \frac{\xi^{(\varsigma - 1)}}{\Gamma_q(\varsigma)} \right)$$

$$+ \Lambda_2^* \left( \frac{\ell_2}{\Gamma_q(\varsigma + \sigma - \varrho + 1)} + \frac{\xi^{(\varsigma - \varrho)}}{\Gamma_q(\varsigma - \varrho + 1)} \right). \tag{20}$$

Consequently, we can declare that $\|\mathcal{G}\hbar\|_{\mathfrak{A}} \leq \hat{\Omega}\vartheta^* \wp(\varepsilon) < \infty$, and this implies uniform boundedness of $\mathcal{G}$ on $\mathfrak{H}$. Next, we ensure the equi-continuity of $\mathcal{G}$. In order to check this, consider $r_1, r_2 \in [0,1]$ such that $r_1 < r_2$ and $\hbar \in \mathfrak{H}$. Then, we get:

$$|(\mathcal{G}\hbar)(r_2) - (\mathcal{G}\hbar)(r_1)| \leq \int_0^{r_1} \frac{[(r_2 - qv)^{(\varsigma-1)} - (r_1 - qv)^{(\varsigma-1)}]}{\Gamma_q(\varsigma)}|\varphi_*(v, \hbar(v))|\, \mathrm{d}_q v$$

$$+ \int_{r_1}^{r_2} \frac{(r_2 - qv)^{(\varsigma-1)}}{\Gamma_q(\varsigma)}|\varphi_*(v, \hbar(v))|\, \mathrm{d}_q v$$

$$+ \ell_3[\Lambda_1(r_2) - \Lambda_1(r_1)] \int_0^1 \frac{(1 - qv)^{(\varsigma+\sigma-2)}}{\Gamma_q(\varsigma + \sigma - 1)}|\varphi_*(v, \hbar(v))|\, \mathrm{d}_q v$$

$$+ [\Lambda_1(r_2) - \Lambda_1(r_1)] \int_0^{\xi} \frac{(\xi - qv)^{(\varsigma-2)}}{\Gamma_q(\varsigma - 1)}|\varphi_*(v, \hbar(v))|\, \mathrm{d}_q v$$

$$+ \ell_2[\Lambda_2(r_2) - \Lambda_2(r_1)] \int_0^1 \frac{(1 - qv)^{(\varsigma+\sigma-\varrho-1)}}{\Gamma_q(\varsigma + \sigma - \varrho)}|\varphi_*(v, \hbar(v))|\, \mathrm{d}_q v$$

$$+ [\Lambda_2(r_2) - \Lambda_2(r_1)] \int_0^{\xi} \frac{(\xi - qv)^{(\varsigma-\varrho-1)}}{\Gamma_q(\varsigma - \varrho)}|\varphi_*(v, \hbar(v))|\, \mathrm{d}_q v.$$

Note that the above inequality's right hand side goes to zero as $r_1 \to r_2$ (independent of $\hbar$). Hence, it is evident that $\|(\mathcal{G}\hbar)(r_2) - (\mathcal{G}\hbar)(r_1)\|_{\mathfrak{A}} \to 0$ as $r_1 \to r_2$, and this confirms that $\mathcal{G}$ is an equi-continuous. Consequently, we conclude that $\mathcal{G}$ is a compact operator on $\mathfrak{H}$ in view of the famous Arzela–Ascoli theorem.

At this point, we will check that $\mathcal{G}$ is condensing operator on $\mathfrak{H}$. By Lemma 3, it is obvious that a countable set $\mathcal{H}_0 = \{\hbar_n\}_{n\geq 1} \subset \mathcal{H}$ exists for each bounded subset $\mathcal{H} \subset \mathfrak{H}$ such that $\mathbb{O}(\mathcal{G}(\mathcal{H})) \leq 2\mathbb{O}(\mathcal{G}(\mathcal{H}_0))$ holds. Hence, in the light of Lemmas 2, 4 and 5, the following is obtained

$$\mathbb{O}(\mathcal{G}(\mathcal{H})(r)) \leq 2\mathbb{O}(\mathcal{G}(\{\hbar_n\}_{n\geq 1}))$$

$$\leq 2 \int_0^r \frac{(r - qv)^{(\varsigma-1)}}{\Gamma_q(\varsigma)}\mathbb{O}(\varphi_*(v, \{\hbar_n(v)\}_{n\geq 1}))\, \mathrm{d}_q v$$

$$+ \frac{2\ell_1}{|\delta_1|} \int_0^1 \frac{(1 - qv)^{(\varsigma+\sigma-1)}}{\Gamma_q(\varsigma + \sigma)}\mathbb{O}(\varphi_*(v, \{\hbar_n(v)\}_{n\geq 1}))\, \mathrm{d}_q v$$

$$+ \frac{2}{|\delta_1|} \int_0^{\xi} \frac{(\xi - qv)^{(\varsigma-1)}}{\Gamma_q(\varsigma)}\mathbb{O}(\varphi_*(v, \{\hbar_n(v)\}_{n\geq 1}))\, \mathrm{d}_q v$$

$$+ 2\ell_3\Lambda_1(r) \int_0^1 \frac{(1 - qv)^{(\varsigma+\sigma-2)}}{\Gamma_q(\varsigma + \sigma - 1)}\mathbb{O}(\varphi_*(v, \{\hbar_n(v)\}_{n\geq 1}))\, \mathrm{d}_q v$$

$$+ 2\Lambda_1(r) \int_0^{\xi} \frac{(\xi - qv)^{(\varsigma-2)}}{\Gamma_q(\varsigma - 1)}\mathbb{O}(\varphi_*(v, \{\hbar_n(v)\}_{n\geq 1}))\, \mathrm{d}_q v$$

$$+ 2\ell_2\Lambda_2(r) \int_0^1 \frac{(1 - qv)^{(\varsigma+\sigma-\varrho-1)}}{\Gamma_q(\varsigma + \sigma - \varrho)}\mathbb{O}(\varphi_*(v, \{\hbar_n(v)\}_{n\geq 1}))\, \mathrm{d}_q v$$

$$+ 2\Lambda_2(r) \int_0^{\xi} \frac{(\xi - qv)^{(\varsigma-\varrho-1)}}{\Gamma_q(\varsigma - \varrho)}\mathbb{O}(\varphi_*(v, \{\hbar_n(v)\}_{n\geq 1}))\, \mathrm{d}_q v$$

$$\leq 4 \int_0^r \frac{(r - qv)^{(\varsigma-1)}}{\Gamma_q(\varsigma)} m_{\varphi_*}(v) \mathbb{O}(\{\hbar_n(v)\}_{n\geq1}) \, \mathrm{d}_q v$$

$$+ \frac{4\ell_1}{|\delta_1|} \int_0^1 \frac{(1 - qv)^{(\varsigma+\sigma-1)}}{\Gamma_q(\varsigma+\sigma)} m_{\varphi_*}(v) \mathbb{O}(\{\hbar_n(v)\}_{n\geq1}) \, \mathrm{d}_q v$$

$$+ \frac{4}{|\delta_1|} \int_0^\xi \frac{(\xi - qv)^{(\varsigma-1)}}{\Gamma_q(\varsigma)} m_{\varphi_*}(v) \mathbb{O}(\{\hbar_n(v)\}_{n\geq1}) \, \mathrm{d}_q v$$

$$+ 4\ell_3 \Lambda_1(r) \int_0^1 \frac{(1 - qv)^{(\varsigma+\sigma-2)}}{\Gamma_q(\varsigma+\sigma-1)} m_{\varphi_*}(v) \mathbb{O}(\{\hbar_n(v)\}_{n\geq1}) \, \mathrm{d}_q v$$

$$+ 4\Lambda_1(r) \int_0^\xi \frac{(\xi - qv)^{(\varsigma-2)}}{\Gamma_q(\varsigma-1)} m_{\varphi_*}(v) \mathbb{O}(\{\hbar_n(v)\}_{n\geq1}) \, \mathrm{d}_q v$$

$$+ 4\ell_2 \Lambda_2(r) \int_0^1 \frac{(1 - qv)^{(\varsigma+\sigma-\varrho-1)}}{\Gamma_q(\varsigma+\sigma-\varrho)} m_{\varphi_*}(v) \mathbb{O}(\{\hbar_n(v)\}_{n\geq1}) \, \mathrm{d}_q v$$

$$+ 4\Lambda_2(r) \int_0^\xi \frac{(\xi - qv)^{(\varsigma-\varrho-1)}}{\Gamma_q(\varsigma-\varrho)} m_{\varphi_*}(v) \mathbb{O}(\{\hbar_n(v)\}_{n\geq1}) \, \mathrm{d}_q v$$

$$\leq 4\tilde{m}_{\varphi_*} \mathbb{O}(\mathcal{H}) \int_0^r \frac{(r - qv)^{(\varsigma-1)}}{\Gamma_q(\varsigma)} \, \mathrm{d}_q v$$

$$+ \frac{4\ell_1 \tilde{m}_{\varphi_*} \mathbb{O}(\mathcal{H})}{|\delta_1|} \int_0^1 \frac{(1 - qv)^{(\varsigma+\sigma-1)}}{\Gamma_q(\varsigma+\sigma)} \, \mathrm{d}_q v + \frac{4\tilde{m}_{\varphi_*} \mathbb{O}(\mathcal{H})}{|\delta_1|} \int_0^\xi \frac{(\xi - qv)^{(\varsigma-1)}}{\Gamma_q(\varsigma)} \, \mathrm{d}_q v$$

$$+ 4\ell_3 \Lambda_1^* \tilde{m}_{\varphi_*} \mathbb{O}(\mathcal{H}) \int_0^1 \frac{(1 - qv)^{(\varsigma+\sigma-2)}}{\Gamma_q(\varsigma+\sigma-1)} \, \mathrm{d}_q v$$

$$+ 4\Lambda_1^* \tilde{m}_{\varphi_*} \mathbb{O}(\mathcal{H}) \int_0^\xi \frac{(\xi - qv)^{(\varsigma-2)}}{\Gamma_q(\varsigma-1)} \, \mathrm{d}_q v$$

$$+ 4\ell_2 \Lambda_2^* \tilde{m}_{\varphi_*} \mathbb{O}(\mathcal{H}) \int_0^1 \frac{(1 - qv)^{(\varsigma+\sigma-\varrho-1)}}{\Gamma_q(\varsigma+\sigma-\varrho)} \, \mathrm{d}_q v$$

$$+ 4\Lambda_2^* \tilde{m}_{\varphi_*} \mathbb{O}(\mathcal{H}) \int_0^\xi \frac{(\xi - qv)^{(\varsigma-\varrho-1)}}{\Gamma_q(\varsigma-\varrho)} \, \mathrm{d}_q v$$

$$\leq \frac{4\tilde{m}_{\varphi_*} \mathbb{O}(\mathcal{H})}{\Gamma_q(\varsigma+1)} + \frac{4\ell_1 \tilde{m}_{\varphi_*} \mathbb{O}(\mathcal{H})}{|\delta_1| \Gamma_q(\varsigma+\sigma+1)} + \frac{4\xi^{(\varsigma)} \tilde{m}_{\varphi_*} \mathbb{O}(\mathcal{H})}{|\delta_1| \Gamma_q(\varsigma+1)} + \frac{4\ell_3 \Lambda_1^* \tilde{m}_{\varphi_*} \mathbb{O}(\mathcal{H})}{\Gamma_q(\varsigma+\sigma)}$$

$$+ \frac{4\xi^{(\varsigma-1)} \Lambda_1^* \tilde{m}_{\varphi_*} \mathbb{O}(\mathcal{H})}{\Gamma_q(\varsigma)} + \frac{4\ell_2 \Lambda_2^* \tilde{m}_{\varphi_*} \mathbb{O}(\mathcal{H})}{\Gamma_q(\varsigma+\sigma-\varrho+1)} + \frac{4\xi^{(\varsigma-\varrho)} \Lambda_2^* \tilde{m}_{\varphi_*} \mathbb{O}(\mathcal{H})}{\Gamma_q(\varsigma-\varrho+1)}.$$

Hence,

$$\mathbb{O}(\mathcal{G}(\mathcal{H})) \leq 4 \left[ \frac{\tilde{m}_{\varphi_*}}{\Gamma_q(\varsigma+1)} + \frac{\tilde{m}_{\varphi_*}}{|\delta_1|} \left( \frac{\ell_1}{\Gamma_q(\varsigma+\sigma+1)} + \frac{\xi^{(\varsigma)}}{\Gamma_q(\varsigma+1)} \right) \right.$$

$$+ \tilde{m}_{\varphi_*} \Lambda_1^* \left( \frac{\ell_3}{\Gamma_q(\varsigma+\sigma)} + \frac{\xi^{(\varsigma-1)}}{\Gamma_q(\varsigma)} \right)$$

$$\left. + \tilde{m}_{\varphi_*} \Lambda_2^* \left( \frac{\ell_2}{\Gamma_q(\varsigma+\sigma-\varrho+1)} + \frac{\xi^{(\varsigma-\varrho)}}{\Gamma_q(\varsigma-\varrho+1)} \right) \right] \mathbb{O}(\mathcal{H}).$$

By applying condition (16), we get $\mathbb{O}(\mathcal{G}(\mathcal{H})) < \mathbb{O}(\mathcal{H})$. This clearly implies that $\mathcal{G}$ is condensing operator on $\mathfrak{H}$. Ultimately, by employing Theorem 1, we can infer that the map $\mathcal{G}$ possesses one fixed point leastwise in $\mathfrak{H}$. Thus, it is found at least one solution for the supposed quantum-integro-difference FBVP (1) and finally the proof process is terminated. $\square$

Now, we set up an existence criterion for the given fractional quantum inclusion BVP (2). The inclusion problem's solution (2) is determined by an absolutely continuous function $\hbar : [0,1] \to \mathbb{R}$ whenever it satisfies the given fractional quantum integro-difference conditions, and a function $\mathfrak{z} \in \mathcal{L}^1([0,1], \mathbb{R})$ exists such that the inclusion $\mathfrak{z}(r) \in \mathbb{T}_*(r, \hbar(r))$ holds for almost all $r \in [0,1]$, and we have:

$$
\hbar(r) = \int_0^r \frac{(r-qv)^{(\varsigma-1)}}{\Gamma_q(\varsigma)} \mathfrak{z}(v)\, \mathrm{d}_q v + \frac{\ell_1}{\delta_1} \int_0^1 \frac{(1-qv)^{(\varsigma+\sigma-1)}}{\Gamma_q(\varsigma+\sigma)} \mathfrak{z}(v)\, \mathrm{d}_q v
$$

$$
- \frac{1}{\delta_1} \int_0^\xi \frac{(\xi-qv)^{(\varsigma-1)}}{\Gamma_q(\varsigma)} \mathfrak{z}(v)\, \mathrm{d}_q v
$$

$$
+ \ell_3 \Lambda_1(r) \int_0^1 \frac{(1-qv)^{(\varsigma+\sigma-2)}}{\Gamma_q(\varsigma+\sigma-1)} \mathfrak{z}(v)\, \mathrm{d}_q v - \Lambda_1(r) \int_0^\xi \frac{(\xi-qv)^{(\varsigma-2)}}{\Gamma_q(\varsigma-1)} \mathfrak{z}(v)\, \mathrm{d}_q v
$$

$$
+ \ell_2 \Lambda_2(r) \int_0^1 \frac{(1-qv)^{(\varsigma+\sigma-\varrho-1)}}{\Gamma_q(\varsigma+\sigma-\varrho)} \mathfrak{z}(v)\, \mathrm{d}_q v - \Lambda_2(r) \int_0^\xi \frac{(\xi-qv)^{(\varsigma-\varrho-1)}}{\Gamma_q(\varsigma-\varrho)} \mathfrak{z}(v)\, \mathrm{d}_q v,
$$

for each $r \in [0,1]$. Let $\mathfrak{S}_{\mathbb{T}_*,\hbar}$ represents the collection of all selections of $\mathbb{T}_*$ for each $\hbar \in \mathfrak{A}$ and is defined as

$$
\mathfrak{S}_{\mathbb{T}_*,\hbar} = \{\mathfrak{z} \in \mathcal{L}^1([0,1]) : \mathfrak{z}(r) \in \mathbb{T}_*(r, \hbar(r)) \text{ for almost all } r \in [0,1]\}.
$$

Construct a multi-valued map $\mathcal{J} : \mathfrak{A} \to \mathbb{P}(\mathfrak{A})$ which is defined as

$$
\mathcal{J}(\hbar) = \{\mathfrak{h} \in \mathfrak{A} : \mathfrak{h}(r) = \varpi(r)\}, \tag{21}
$$

where

$$
\varpi(r) = \int_0^r \frac{(r-qv)^{(\varsigma-1)}}{\Gamma_q(\varsigma)} \mathfrak{z}(v)\, \mathrm{d}_q v + \frac{\ell_1}{\delta_1} \int_0^1 \frac{(1-qv)^{(\varsigma+\sigma-1)}}{\Gamma_q(\varsigma+\sigma)} \mathfrak{z}(v)\, \mathrm{d}_q v
$$

$$
- \frac{1}{\delta_1} \int_0^\xi \frac{(\xi-qv)^{(\varsigma-1)}}{\Gamma_q(\varsigma)} \mathfrak{z}(v)\, \mathrm{d}_q v
$$

$$
+ \ell_3 \Lambda_1(r) \int_0^1 \frac{(1-qv)^{(\varsigma+\sigma-2)}}{\Gamma_q(\varsigma+\sigma-1)} \mathfrak{z}(v)\, \mathrm{d}_q v - \Lambda_1(r) \int_0^\xi \frac{(\xi-qv)^{(\varsigma-2)}}{\Gamma_q(\varsigma-1)} \mathfrak{z}(v)\, \mathrm{d}_q v
$$

$$
+ \ell_2 \Lambda_2(r) \int_0^1 \frac{(1-qv)^{(\varsigma+\sigma-\varrho-1)}}{\Gamma_q(\varsigma+\sigma-\varrho)} \mathfrak{z}(v)\, \mathrm{d}_q v
$$

$$
- \Lambda_2(r) \int_0^\xi \frac{(\xi-qv)^{(\varsigma-\varrho-1)}}{\Gamma_q(\varsigma-\varrho)} \mathfrak{z}(v)\, \mathrm{d}_q v, \qquad \mathfrak{z} \in \mathfrak{S}_{\mathbb{T}_*,\hbar}.
$$

**Theorem 4.** *Let* $\mathbb{T}_* : [0,1] \times \mathfrak{A} \to \mathbb{P}_{cm}(\mathfrak{A})$ *be a multi-valued map. Suppose that*

$(\mathcal{A}_1)$ *an increasing u.s.c map* $\psi : [0,\infty) \to [0,\infty)$ *exists such that* $\liminf_{r\to\infty}(r - \psi(r)) > 0$, *and* $\psi(r) < r$ *for every* $r > 0$;

$(\mathcal{A}_2)$ $\mathbb{T}_* : [0,1] \times \mathfrak{A} \to \mathbb{P}_{cm}(\mathfrak{A})$ *is integrable and bounded and* $\mathbb{T}_*(\cdot, \hbar) : [0,1] \to \mathbb{P}_{cm}(\mathfrak{A})$ *is measurable for every* $\hbar \in \mathfrak{A}$;

($\mathcal{A}_3$)  $\zeta \in \mathcal{C}([0,1], [0,\infty))$ *exists subject to*

$$\mathbb{H}_d(\mathbb{T}_*(r, \hbar_1(r)), \mathbb{T}_*(r, \hbar_2(r))) \leq \zeta(r)\psi(|\hbar_1(r) - \hbar_2(r)|)\,\frac{1}{Q},$$

*for each $r \in [0,1]$ and $\hbar_1, \hbar_2 \in \mathfrak{A}$, where $\sup_{r \in [0,1]} |\zeta(r)| = \|\zeta\|$ and*

$$Q = \left[ \frac{1}{\Gamma_q(\varsigma+1)} + \frac{1}{|\delta_1|}\left(\frac{\ell_1}{\Gamma_q(\varsigma+\sigma+1)} + \frac{\xi^{(\varsigma)}}{\Gamma_q(\varsigma+1)}\right) + \Lambda_1^*\left(\frac{\ell_3}{\Gamma_q(\varsigma+\sigma)} + \frac{\xi^{(\varsigma-1)}}{\Gamma_q(\varsigma)}\right) \right.$$

$$\left. + \Lambda_2^*\left(\frac{\ell_2}{\Gamma_q(\varsigma+\sigma-\varrho+1)} + \frac{\xi^{(\varsigma-\varrho)}}{\Gamma_q(\varsigma-\varrho+1)}\right) \right] \|\zeta\|; \tag{22}$$

($\mathcal{A}_4$) *the multi-valued map $\mathcal{J} : \mathfrak{A} \to \mathbb{P}(\mathfrak{A})$ formulated in* (21) *satisfies approximate endpoint criterion.*

*Then, a solution is found for the given quantum-difference inclusion FBVP* (2).

**Proof.** We are going to determine that an endpoint exists for the multifunction $\mathcal{J} : \mathfrak{A} \to \mathbb{P}(\mathfrak{A})$ given by (21). Since the map $r \to \mathbb{T}_*(r, \hbar(r))$ is measurable and closed-valued set-valued mappingl therefore, it has a measurable selection. As a result, $\mathfrak{S}_{\mathbb{T}_*, \hbar} \neq \varnothing$. Firstly, we show that $\mathcal{J}(\hbar)$ is closed for every $\hbar \in \mathfrak{A}$. Consider the sequence $\{\hbar_n\}_{n\geq 1}$ in $\mathcal{J}(\hbar)$ such that $\hbar_n$ converges to $\hbar$. For each n, there exists $\mathfrak{z}_n \in \mathfrak{S}_{\mathbb{T}_*, \hbar}$ such that

$$\hbar_n(r) = \int_0^r \frac{(r-qv)^{(\varsigma-1)}}{\Gamma_q(\varsigma)}\mathfrak{z}_n(v)\,\mathrm{d}_q v + \frac{\ell_1}{\delta_1}\int_0^1 \frac{(1-qv)^{(\varsigma+\sigma-1)}}{\Gamma_q(\varsigma+\sigma)}\mathfrak{z}_n(v)\,\mathrm{d}_q v$$

$$- \frac{1}{\delta_1}\int_0^\xi \frac{(\xi-qv)^{(\varsigma-1)}}{\Gamma_q(\varsigma)}\mathfrak{z}_n(v)\,\mathrm{d}_q v$$

$$+ \ell_3 \Lambda_1(r)\int_0^1 \frac{(1-qv)^{(\varsigma+\sigma-2)}}{\Gamma_q(\varsigma+\sigma-1)}\mathfrak{z}_n(v)\,\mathrm{d}_q v - \Lambda_1(r)\int_0^\xi \frac{(\xi-qv)^{(\varsigma-2)}}{\Gamma_q(\varsigma-1)}\mathfrak{z}_n(v)\,\mathrm{d}_q v$$

$$+ \ell_2 \Lambda_2(r)\int_0^1 \frac{(1-qv)^{(\varsigma+\sigma-\varrho-1)}}{\Gamma_q(\varsigma+\sigma-\varrho)}\mathfrak{z}_n(v)\,\mathrm{d}_q v - \Lambda_2(r)\int_0^\xi \frac{(\xi-qv)^{(\varsigma-\varrho-1)}}{\Gamma_q(\varsigma-\varrho)}\mathfrak{z}_n(v)\,\mathrm{d}_q v,$$

for almost all $r \in [0,1]$. Since the multi-valued function $\mathbb{T}_*$ is compact, we have a subsequence $\{\mathfrak{z}_n\}_{n\geq 1}$ converging to $\mathfrak{z} \in \mathcal{L}^1([0,1])$. Thus, $\mathfrak{z} \in \mathfrak{S}_{\mathbb{T}_*, \hbar}$ and

$$\lim_{n\to\infty} \hbar_n(r) = \int_0^r \frac{(r-qv)^{(\varsigma-1)}}{\Gamma_q(\varsigma)}\mathfrak{z}(v)\,\mathrm{d}_q v$$

$$+ \frac{\ell_1}{\delta_1}\int_0^1 \frac{(1-qv)^{(\varsigma+\sigma-1)}}{\Gamma_q(\varsigma+\sigma)}\mathfrak{z}(v)\,\mathrm{d}_q v - \frac{1}{\delta_1}\int_0^\xi \frac{(\xi-qv)^{(\varsigma-1)}}{\Gamma_q(\varsigma)}\mathfrak{z}(v)\,\mathrm{d}_q v$$

$$+ \ell_3 \Lambda_1(r)\int_0^1 \frac{(1-qv)^{(\varsigma+\sigma-2)}}{\Gamma_q(\varsigma+\sigma-1)}\mathfrak{z}(v)\,\mathrm{d}_q v - \Lambda_1(r)\int_0^\xi \frac{(\xi-qv)^{(\varsigma-2)}}{\Gamma_q(\varsigma-1)}\mathfrak{z}(v)\,\mathrm{d}_q v$$

$$+ \ell_2 \Lambda_2(r)\int_0^1 \frac{(1-qv)^{(\varsigma+\sigma-\varrho-1)}}{\Gamma_q(\varsigma+\sigma-\varrho)}\mathfrak{z}(v)\,\mathrm{d}_q v - \Lambda_2(r)\int_0^\xi \frac{(\xi-qv)^{(\varsigma-\varrho-1)}}{\Gamma_q(\varsigma-\varrho)}\mathfrak{z}(v)\,\mathrm{d}_q v$$

$$= \hbar(r),$$

for almost all $r \in [0,1]$. This indicates that $\hbar \in \mathcal{J}$ and therefore, $\mathcal{J}$ is closed-valued. Since $\mathbb{T}_*$ is compact multi-valued function, it is simple to check that $\mathcal{J}(\hbar)$ is bounded for all

$\hbar \in \mathfrak{A}$. At last, we prove that $\mathbb{H}_d(\mathcal{J}(\hbar_1), \mathcal{J}(\hbar_2)) \leq \psi(\|\hbar_1 - \hbar_2\|)$ holds. Let $\hbar_1, \hbar_2 \in \mathfrak{A}$ and $\tau_1 \in \mathcal{J}(\hbar_2)$. Select $\mathfrak{z}_1 \in \mathfrak{S}_{\mathbb{T}_*, \hbar}$ such that

$$\tau_1(r) = \int_0^r \frac{(r - qv)^{(\varsigma-1)}}{\Gamma_q(\varsigma)} \mathfrak{z}_1(v) \, d_q v$$

$$+ \frac{\ell_1}{\delta_1} \int_0^1 \frac{(1 - qv)^{(\varsigma+\sigma-1)}}{\Gamma_q(\varsigma + \sigma)} \mathfrak{z}_1(v) \, d_q v - \frac{1}{\delta_1} \int_0^\xi \frac{(\xi - qv)^{(\varsigma-1)}}{\Gamma_q(\varsigma)} \mathfrak{z}_1(v) \, d_q v$$

$$+ \ell_3 \Lambda_1(r) \int_0^1 \frac{(1 - qv)^{(\varsigma+\sigma-2)}}{\Gamma_q(\varsigma + \sigma - 1)} \mathfrak{z}_1(v) \, d_q v - \Lambda_1(r) \int_0^\xi \frac{(\xi - qv)^{(\varsigma-2)}}{\Gamma_q(\varsigma - 1)} \mathfrak{z}_1(v) \, d_q v$$

$$+ \ell_2 \Lambda_2(r) \int_0^1 \frac{(1 - qv)^{(\varsigma+\sigma-\varrho-1)}}{\Gamma_q(\varsigma + \sigma - \varrho)} \mathfrak{z}_1(v) \, d_q v - \Lambda_2(r) \int_0^\xi \frac{(\xi - qv)^{(\varsigma-\varrho-1)}}{\Gamma_q(\varsigma - \varrho)} \mathfrak{z}_1(v) \, d_q v,$$

for all $r \in [0, 1]$. Since

$$\mathbb{H}_d(\mathbb{T}_*(r, \hbar_1(r)), \mathbb{T}_*(r, \hbar_2(r))) \leq \zeta(r) \psi(|\hbar_1(r) - \hbar_2(r)|) \frac{1}{\mathcal{Q}}$$

for each $r \in [0, 1]$, so there exists $\mathfrak{z}^* \in \mathbb{T}_*(r, \hbar_1(r))$ such that

$$|\mathfrak{z}_1(r) - \mathfrak{z}^*| \leq \zeta(r) \psi(|\hbar_1(r) - \hbar_2(r)|) \frac{1}{\mathcal{Q}},$$

for each $r \in [0, 1]$. Now, the multi-valued map $\mathfrak{X} : [0, 1] \to \mathbb{P}(\mathfrak{A})$ is considered, which is characterized by

$$\mathfrak{X}(r) = \left\{ \mathfrak{z}^* \in \mathfrak{A} : |\mathfrak{z}_1(r) - \mathfrak{z}^*| \leq \zeta(r) \psi(|\hbar_1(r) - \hbar_2(r)|) \frac{1}{\mathcal{Q}} \right\}.$$

Since $\mathfrak{z}_1$ and $\eta = \zeta(\psi(\hbar_1 - \hbar_2)) \frac{1}{\mathcal{Q}}$ are measurable, so it is obvious that the multifunction $\mathfrak{X} \cap \mathbb{T}_*(\cdot, \hbar(\cdot))$ is measurable. Now, select $\mathfrak{z}_2(r) \in \mathbb{T}_*(r, \hbar(r))$ such that

$$|\mathfrak{z}_1(r) - \mathfrak{z}_2(r)| \leq \zeta(r)(\psi(|\hbar_1(r) - \hbar_2(r)|)) \frac{1}{\mathcal{Q}},$$

for all $r \in [0, 1]$. Choose $\tau_2 \in \mathcal{J}(\hbar_1)$ such that

$$\tau_2(r) = \int_0^r \frac{(r - qv)^{(\varsigma-1)}}{\Gamma_q(\varsigma)} \mathfrak{z}_2(v) \, d_q v$$

$$+ \frac{\ell_1}{\delta_1} \int_0^1 \frac{(1 - qv)^{(\varsigma+\sigma-1)}}{\Gamma_q(\varsigma + \sigma)} \mathfrak{z}_2(v) \, d_q v - \frac{1}{\delta_1} \int_0^\xi \frac{(\xi - qv)^{(\varsigma-1)}}{\Gamma_q(\varsigma)} \mathfrak{z}_2(v) \, d_q v$$

$$+ \ell_3 \Lambda_1(r) \int_0^1 \frac{(1 - qv)^{(\varsigma+\sigma-2)}}{\Gamma_q(\varsigma + \sigma - 1)} \mathfrak{z}_2(v) \, d_q v - \Lambda_1(r) \int_0^\xi \frac{(\xi - qv)^{(\varsigma-2)}}{\Gamma_q(\varsigma - 1)} \mathfrak{z}_2(v) \, d_q v$$

$$+ \ell_2 \Lambda_2(r) \int_0^1 \frac{(1 - qv)^{(\varsigma+\sigma-\varrho-1)}}{\Gamma_q(\varsigma + \sigma - \varrho)} \mathfrak{z}_2(v) \, d_q v - \Lambda_2(r) \int_0^\xi \frac{(\xi - qv)^{(\varsigma-\varrho-1)}}{\Gamma_q(\varsigma - \varrho)} \mathfrak{z}_2(v) \, d_q v,$$

for any $r \in [0, 1]$. Then, we get

$$|\tau_1(r) - \tau_2(r)| \leq \int_0^r \frac{(r - qv)^{(\varsigma-1)}}{\Gamma_q(\varsigma)} |\mathfrak{z}_1(v) - \mathfrak{z}_2(v)| \, d_q v$$

$$+ \frac{\ell_1}{|\delta_1|} \int_0^1 \frac{(1-qv)^{(\varsigma+\sigma-1)}}{\Gamma_q(\varsigma+\sigma)} |\mathfrak{z}_1(v) - \mathfrak{z}_2(v)| \, \mathrm{d}_q v$$

$$+ \frac{1}{|\delta_1|} \int_0^\xi \frac{(\xi-qv)^{(\varsigma-1)}}{\Gamma_q(\varsigma)} |\mathfrak{z}_1(v) - \mathfrak{z}_2(v)| \, \mathrm{d}_q v$$

$$+ \ell_3 |\Lambda_1(r)| \int_0^1 \frac{(1-qv)^{(\varsigma+\sigma-2)}}{\Gamma_q(\varsigma+\sigma-1)} |\mathfrak{z}_1(v) - \mathfrak{z}_2(v)| \, \mathrm{d}_q v$$

$$+ |\Lambda_1(r)| \int_0^\xi \frac{(\xi-qv)^{(\varsigma-2)}}{\Gamma_q(\varsigma-1)} |\mathfrak{z}_1(v) - \mathfrak{z}_2(v)| \, \mathrm{d}_q v$$

$$+ \ell_2 |\Lambda_2(r)| \int_0^1 \frac{(1-qv)^{(\varsigma+\sigma-\varrho-1)}}{\Gamma_q(\varsigma+\sigma-\varrho)} |\mathfrak{z}_1(v) - \mathfrak{z}_2(v)| \, \mathrm{d}_q v$$

$$+ |\Lambda_2(r)| \int_0^\xi \frac{(\xi-qv)^{(\varsigma-\varrho-1)}}{\Gamma_q(\varsigma-\varrho)} |\mathfrak{z}_1(v) - \mathfrak{z}_2(v)| \, \mathrm{d}_q v$$

$$\leq \frac{1}{\Gamma_q(\varsigma+1)} \|\zeta\| \psi(\|\hbar_1 - \hbar_2\|) \frac{1}{\mathcal{Q}}$$

$$+ \frac{\ell_1}{|\delta_1|\Gamma_q(\varsigma+\sigma+1)} \|\zeta\| \psi(\|\hbar_1 - \hbar_2\|) \frac{1}{\mathcal{Q}} + \frac{\xi^{(\varsigma)}}{|\delta_1|\Gamma_q(\varsigma+1)} \|\zeta\| \psi(\|\hbar_1 - \hbar_2\|) \frac{1}{\mathcal{Q}}$$

$$+ \frac{\ell_3 \Lambda_1^*}{\Gamma_q(\varsigma+\sigma)} \|\zeta\| \psi(\|\hbar_1 - \hbar_2\|) \frac{1}{\mathcal{Q}} + \frac{\Lambda_1^* \xi^{(\varsigma-1)}}{\Gamma_q(\varsigma)} \|\zeta\| \psi(\|\hbar_1 - \hbar_2\|) \frac{1}{\mathcal{Q}}$$

$$+ \frac{\ell_2 \Lambda_2^*}{\Gamma_q(\varsigma+\sigma-\varrho+1)} \|\zeta\| \psi(\|\hbar_1 - \hbar_2\|) \frac{1}{\mathcal{Q}} + \frac{\Lambda_2^* \xi^{(\varsigma-\varrho)}}{\Gamma_q(\varsigma-\varrho+1)} \|\zeta\| \psi(\|\hbar_1 - \hbar_2\|) \frac{1}{\mathcal{Q}}$$

$$= \left[ \frac{1}{\Gamma_q(\varsigma+1)} + \frac{1}{|\delta_1|} \left( \frac{\ell_1}{\Gamma_q(\varsigma+\sigma+1)} + \frac{\xi^{(\varsigma)}}{\Gamma_q(\varsigma+1)} \right) \right.$$

$$+ \Lambda_1^* \left( \frac{\ell_3}{\Gamma_q(\varsigma+\sigma)} + \frac{\xi^{(\varsigma-1)}}{\Gamma_q(\varsigma)} \right)$$

$$\left. + \Lambda_2^* \left( \frac{\ell_2}{\Gamma_q(\varsigma+\sigma-\varrho+1)} + \frac{\xi^{(\varsigma-\varrho)}}{\Gamma_q(\varsigma-\varrho+1)} \right) \right] \|\zeta\| \psi(\|\hbar_1 - \hbar_2\|) \frac{1}{\mathcal{Q}}$$

$$= \mathcal{Q} \psi(\|\hbar_1 - \hbar_2\|) \frac{1}{\mathcal{Q}}$$

$$= \psi(\|\hbar_1 - \hbar_2\|).$$

Thus, we get $\|\tau_1 - \tau_2\| \leq \psi(\|\hbar_1 - \hbar_2\|)$. Hence, $\mathbb{H}_d(\mathcal{J}(\hbar_1), \mathcal{J}(\hbar_1)) \leq \psi(\|\hbar_1 - \hbar_2\|)$ for each $\hbar_1, \hbar_2 \in \mathfrak{A}$. By utilizing $(\mathcal{A}_4)$, we realize that $\mathcal{J}$ has an approximate endpoint criterion. Now by employing Theorem 2, a member $\hbar^* \in \mathfrak{A}$ exists such that $\mathcal{J}(\hbar^*) = \{\hbar^*\}$. This indicates that $\hbar^*$ is the solution of the fractional quantum-difference inclusion problem (2), hence, our proof is finally completed. □

## 4. Numerical Examples

This section provides some interesting numerical examples to apply and validate our results in this research work.

**Example 1.** *Consider the following Caputo quantum-difference FBVP:*

$$
\begin{cases}
{}^{C}_{0.5}\mathfrak{D}^{2.5}_{0^+}\hbar(r) = \dfrac{3r+1}{8000e^{-r}}\sin(\hbar(r)), \\[2mm]
\hbar(0) + \hbar(0.25) = (0.1)\,{}^{R}_{0.5}\mathfrak{J}^{0.75}_{0^+}\hbar(1), \\[2mm]
{}^{C}_{0.5}\mathfrak{D}^{1.5}_{0^+}\hbar(0) + {}^{C}_{0.5}\mathfrak{D}^{1.5}_{0^+}\hbar(0.25) = (0.2)\,{}^{R}_{0.5}\mathfrak{J}^{0.75}_{0^+}\big[{}^{C}_{0.5}\mathfrak{D}^{1.5}_{0^+}\hbar\big](1), \\[2mm]
{}^{C}_{0.5}\mathfrak{D}^{1}_{0^+}\hbar(0) + {}^{C}_{0.5}\mathfrak{D}^{1}_{0^+}\hbar(0.25) = (0.3)\,{}^{R}_{0.5}\mathfrak{J}^{0.75}_{0^+}\big[{}^{C}_{0.5}\mathfrak{D}^{1}_{0^+}\hbar\big](1),
\end{cases}
\tag{23}
$$

*such that* $q = 0.5$, $\ell_1 = 0.1$, $\varsigma = 2.5$, $\xi = 0.25$, $\ell_2 = 0.2$, $\sigma = 0.75$, $\varrho = 1.5$, $\ell_3 = 0.3$ *and* $r \in [0, 1]$. *Furthermore, we consider a continuous function* $\varphi_*(r, \hbar(r)) : [0,1] \times \mathbb{R} \to \mathbb{R}$ *constructed as:*

$$
\varphi_*(r, \hbar(r)) = \frac{3r+1}{8000e^{-r}}\sin(\hbar(r)).
$$

*The graph of this function is shown in Figure 2.*

**Figure 2.** Graph of the function $\varphi_*(r, \hbar)$ on $[0, 1] \times [0, 50]$.

*Then, for each* $\hbar \in \mathbb{R}$, *we have:*

$$
|\varphi_*(r, \hbar(r))| = \frac{3r+1}{8000e^{-r}}|\sin(\hbar(r))| \le \frac{3r+1}{8000e^{-r}} = \vartheta(r)\wp(\|\hbar\|_{\mathbb{R}}),
$$

*where* $\vartheta : [0,1] \to \mathbb{R}^{>0}$ *is a continuous function defined by* $\vartheta(r) = \frac{3r+1}{8000e^{-r}}$ *and* $\wp : \mathbb{R}^{\ge 0} \to \mathbb{R}^{>0}$ *is nondecreasing and continuous via* $\wp(\|\hbar\|_{\mathbb{R}}) = 1$. *Now, for any* $\hbar_1, \hbar_2 \in \mathbb{R}$, *we can write:*

$$
|\varphi_*(r, \hbar_1(r)) - \varphi_*(r, \hbar_2(r))| = \frac{3r+1}{8000e^{-r}}|\sin(\hbar_1(r)) - \sin(\hbar_2(r))|
$$

$$
\le \frac{3r+1}{8000e^{-r}}|\hbar_1(r) - \hbar_2(r)|.
$$

*Hence, for any bounded set $\mathcal{H}$ contained in $\mathbb{R}$, we reach*

$$\mathbb{O}(\varphi_*(r, \mathcal{H})) \le \frac{3r+1}{8000e^{-r}}\mathbb{O}(\mathcal{H}) := m_{\varphi_*}\mathbb{O}(\mathcal{H}).$$

*We compute $\tilde{m}_{\varphi_*} = \sup_{r \in [0,1]}|m_{\varphi_*}| \simeq 0.001355$. Then, by taking into account the above calculations and the following inequality, we get*

$$\left[ \frac{\tilde{m}_{\varphi_*}}{\Gamma_q(\varsigma+1)} + \frac{\tilde{m}_{\varphi_*}}{|\delta_1|}\left( \frac{\ell_1}{\Gamma_q(\varsigma+\sigma+1)} + \frac{\xi^{(\varsigma)}}{\Gamma_q(\varsigma+1)} \right) + \tilde{m}_{\varphi_*}\Lambda_1^*\left( \frac{\ell_3}{\Gamma_q(\varsigma+\sigma)} + \frac{\xi^{(\varsigma-1)}}{\Gamma_q(\varsigma)} \right) \right.$$

$$\left. + \tilde{m}_{\varphi_*}\Lambda_2^*\left( \frac{\ell_2}{\Gamma_q(\varsigma+\sigma-\varrho+1)} + \frac{\xi^{(\varsigma-\varrho)}}{\Gamma_q(\varsigma-\varrho+1)} \right) \right] \simeq 0.001741 < 0.25 = \frac{1}{4}.$$

*We figure out that Theorem 3 is settled. As a result, at least one solution exists for Caputo fractional quantum-difference FBVP (23).*

**Example 2.** *Consider the following Caputo fractional quantum-difference inclusion FBVP:*

$$\begin{cases} {}^C_{0.8}\mathfrak{D}^{2.75}_{0^+}\hbar(r) \in \left[ 0, \dfrac{5(r+1)\arctan(\hbar(r))}{256(4+3r^2)} \right], \\[2mm] \hbar(0) + \hbar(0.9) = (0.11)\,{}^R_{0.8}\mathfrak{I}^{0.6}_{0^+}\hbar(1), \\[2mm] {}^C_{0.8}\mathfrak{D}^{1.7}_{0^+}\hbar(0) + {}^C_{0.8}\mathfrak{D}^{1.7}_{0^+}\hbar(0.9) = (0.12)\,{}^R_{0.8}\mathfrak{I}^{0.6}_{0^+}\left[{}^C_{0.8}\mathfrak{D}^{1.7}_{0^+}\hbar\right](1), \\[2mm] {}^C_{0.8}\mathfrak{D}^{1}_{0^+}\hbar(0) + {}^C_{0.8}\mathfrak{D}^{1}_{0^+}\hbar(0.9) = (0.13)\,{}^R_{0.8}\mathfrak{I}^{0.6}_{0^+}\left[{}^C_{0.8}\mathfrak{D}^{1}_{0^+}\hbar\right](1), \end{cases} \quad (24)$$

*where $q = 0.8$, $\varsigma = 2.75$, $\xi = 0.9$, $\ell_1 = 0.11$, $\ell_2 = 0.12$, $\ell_3 = 0.13$, $\sigma = 0.6$, $\varrho = 1.7$, and $r \in [0,1]$. Now, we introduce a multi-valued function $\mathbb{T}_* : [0,1] \times \mathbb{R} \to \mathbb{P}(\mathbb{R})$ as follows:*

$$\mathbb{T}_*(r, \hbar(r)) = \left[ 0, \frac{5(r+1)\arctan(\hbar(r))}{256(4+3r^2)} \right].$$

*Next, we regard $\psi : [0, \infty) \to [0, \infty)$ as increasing upper semi-continuous function defined by $\psi(r) = \frac{r}{4}$ for any $r > 0$. It can easily be noted that $\liminf_{r \to \infty}(r - \psi(r)) > 0$ and $\psi(r) < r$ for each $r > 0$. We select $\zeta \in \mathcal{C}([0,1], [0, \infty))$ formulated by $\zeta(r) = \frac{5(r+1)}{64(4+3r^2)}$. Thus, $\|\zeta\| \simeq 0.0390625$. For any $\hbar, \hbar^* \in \mathbb{R}$, we have:*

$$\mathbb{H}_d(\mathbb{T}_*(r, \hbar(r)) - \mathbb{T}_*(r, \hbar^*(r))) = \frac{5(r+1)}{256(4+3r^2)}|\arctan(\hbar(r)) - \arctan(\hbar^*(r))|$$

$$\le \frac{5(r+1)}{256(4+3r^2)}|\hbar(r) - \hbar^*(r)|$$

$$= \frac{5(r+1)}{64(4+3r^2)}\psi(|\hbar(r) - \hbar^*(r)|)$$

$$\le \zeta(r)\psi(|\hbar(r) - \hbar^*(r)|)\frac{1}{\mathcal{Q}},$$

*where*

$$\mathcal{Q} = \left[ \frac{1}{\Gamma_q(\varsigma+1)} + \frac{1}{|\delta_1|}\left( \frac{\ell_1}{\Gamma_q(\varsigma+\sigma+1)} + \frac{\xi^{(\varsigma)}}{\Gamma_q(\varsigma+1)} \right) + \Lambda_1^*\left( \frac{\ell_3}{\Gamma_q(\varsigma+\sigma)} + \frac{\xi^{(\varsigma-1)}}{\Gamma_q(\varsigma)} \right) \right.$$

$$+ \Lambda_2^* \left( \frac{\ell_2}{\Gamma_q(\varsigma + \sigma - \varrho + 1)} + \frac{\xi^{(\varsigma - \varrho)}}{\Gamma_q(\varsigma - \varrho + 1)} \right) \right] \|\zeta\| \simeq 0.066907.$$

*The graphs of the functions: $\Lambda_1(r)$ and $\Lambda_2(r)$ for $r \in [0,1]$ are shown in Figure 3.*

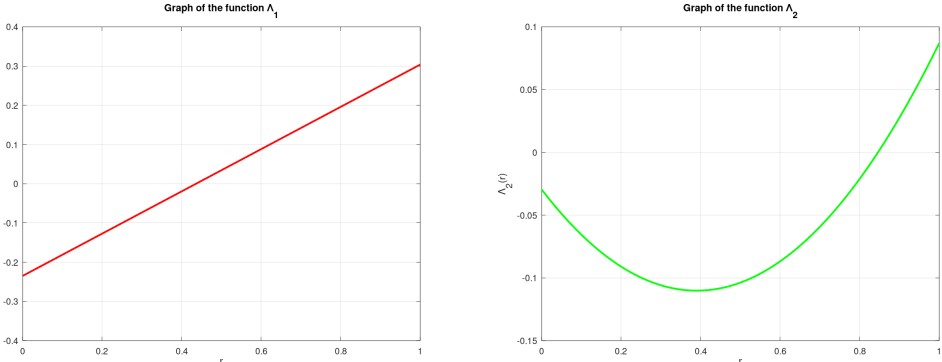

**Figure 3.** Graphs of functions: $\Lambda_1(r)$ and $\Lambda_2(r)$ for $r \in [0,1]$.

*Next, consider the multifunction $\mathcal{J} : \mathfrak{A} \to \mathbb{P}(\mathfrak{A})$ given by:*

$$\mathcal{J}(\hbar) = \{\mathfrak{h} \in \mathfrak{A} : \text{there exists } \mathfrak{z} \in \mathfrak{S}_{\mathbb{T}_*, \hbar} \text{ such that } \mathfrak{h}(r) = \wp(r) \text{ for all } r \in [0,1]\},$$

*where*

$$\wp(r) = \int_0^r \frac{(r - qv)^{(2.75-1)}}{\Gamma_q(2.75)} \mathfrak{z}(v) \, \mathrm{d}_q v + \frac{0.11}{1.8784} \int_0^1 \frac{(1 - qv)^{(2.75+0.6-1)}}{\Gamma_q(2.75 + 0.6)} \mathfrak{z}(v) \, \mathrm{d}_q v$$

$$- \frac{1}{1.8784} \int_0^{0.9} \frac{(0.9 - qv)^{(2.75-1)}}{\Gamma_q(2.75)} \mathfrak{z}(v) \, \mathrm{d}_q v$$

$$+ (0.13) \Lambda_1(r) \int_0^1 \frac{(1 - qv)^{(2.75+0.6-2)}}{\Gamma_q(2.75 + 0.6 - 1)} \mathfrak{z}(v) \, \mathrm{d}_q v - \Lambda_1(r) \int_0^{0.9} \frac{(0.9 - qv)^{(2.75-2)}}{\Gamma_q(2.75 - 1)} \mathfrak{z}(v) \, \mathrm{d}_q v$$

$$+ (0.12) \Lambda_2(r) \int_0^1 \frac{(1 - qv)^{(2.75+0.6-1.7-1)}}{\Gamma_q(2.75 + 0.6 - 1.7)} \mathfrak{z}(v) \, \mathrm{d}_q v$$

$$- \Lambda_2(r) \int_0^{0.9} \frac{(0.9 - qv)^{(2.75-1.7-1)}}{\Gamma_q(2.75 - 1.7)} \mathfrak{z}(v) \, \mathrm{d}_q v,$$

*with $\delta_1 \simeq 1.8784$ and*

$$\Lambda_1(r) = 0.5387r - 0.2348 \quad \text{and} \quad \Lambda_2(r) = 0.53002r^2 - 0.4133r - 0.02959.$$

*Hence, by utilizing Theorem 4, it is found a solution for the quantum-difference inclusion FBVP (24).*

## 5. Conclusions

The proposed nonlinear Caputo quantum-difference FBVP with fractional quantum integro-conditions along with its fractional quantum-difference inclusion BVP has been studied in this work. In this direction, we proved the existence of a solution for the first quantum-difference Equation (1) with the help of some notions in topological degree theory. In other words, we defined a new operator and checked its properties and finally showed that it is a condensing function. The existence of a fixed point for this operator ensured the existence of a solution for the mentioned quantum-difference Equation (1). In the next step, we considered the inclusion version of the above FBVP which had a form as (2). To

arrive at the main purpose this time for confirming the existence of solutions of (2), we used new techniques based on the approximate endpoint property and the existence of endpoints for a newly-defined multifunction. Numerical illustrative examples have been provided to display the validity and potentiality of our main results to be applied in future research works. We recommend that other researchers can study different generalizations of the proposed q-difference-FBVPs by using novel fractional difference-operators such as $(p, q)$-difference ones.

**Author Contributions:** Conceptualization, S.R., A.I., A.H. and S.E.; Formal analysis, S.R., A.I., A.H., F.M., S.E. and M.K. A.K.; Investigation, S.R., A.I., A.H., S.E. and M.K.A.K.; Methodology, S.R., A.I., S.E. and M.K.A.K.; Supervision, S.R., A.I., F.M., S.E. and M.K.A.K.; Validation, A.H., F.M., S.E. and M.K.A.K.; Writing—original draft, S.E.; Writing—review and editing, S.R., A.I., A.H., F.M., S.E. and M.K.A.K. All authors have read and agreed to the published version of the manuscript.

**Funding:** This research received no external funding.

**Institutional Review Board Statement:** Not applicable.

**Informed Consent Statement:** Not applicable.

**Data Availability Statement:** Data sharing not applicable.

**Conflicts of Interest:** The authors declare no conflict of interest.

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
