# Peer review of "Condensing Functions and Approximate Endpoint Criterion for the Existence Analysis of Quantum Integro-Difference FBVPs"

_symmetry, doi:10.3390/sym13030469_

Round 1
Reviewer 1 Report
A quantum boundary value problem with fractional q-integro-difference conditions is investigated. Condensing functions and approximate endpoint criterion are applied. Two numerical examples are provided.
The proof of Proposition 2.1 well specify and clarify about the previous symbols and definition, the proof itself is well explained.
About Theorem 2.3, the using of Lebesgue dominated convergence theorem makes the proof rigorous and well generalized, the same about the Arzela-Ascoli theorem.
From the line 119 authors makes a very shrewd definition respect the choice of functional space.
Numerical examples well reproduces all the theory of the paper in a simple and understandable way.
Main remarks:
- Please avoid the boldface chars for whole single sentences (lines 12, 15, 20 and on).
- I suppose that the section starting in page 1 is the Introduction, if so give the number and name section, or anyway give a name to this part of the paper.
- The first part of the paper could be devoted mainly to the state of the art concerning the involved topic, and could be avoided formulas and definition, which instead could be put in a new section or in the next one into the paper.
- In the first part of the paper it's fundamental the presence of references, here this is almost inexistent.
- The involved Algorithms 1, 2 and 3, consider a first difference approximation which isn't completing challenging respect the state of the art respect this kind of approximation methods. In particular the power functions can't support a consistent convergence respect the asymptotic computation.
- Figure 1: the Caputo derivative approximation must be compared with the exact analytical computation.
- Lemma 1.1 must be studied by considering the regularity properties of the involved function, this is fundamental respect the using of fractional derivatives.
- The approximation result about theorem 2.3 could be more significant if the fraction 1/4 would be replaced by something like 1/ε where ε > 0.
Minor remarks:
- Eq. 1.2: delete the blank space after the first round bracket. Check this kind of misprint alla along the paper (eq. 1.4 for instance, and on).
- Page 6 and 7 are almost completely dedicated to known definition lemmas and theorems, try to make this part more compact.
- Numerical examples: specify the computation integral method used.
Author Response
Dear reviewer
In the attachment you will find the answers to your comments regarding our manuscript. The authors would like to thank you for your involvement in improving this paper.

Reviewer 2 Report
Please see the attached report.

Author Response

(The authors gave the same response as above.)

Round 2
Reviewer 1 Report
I very appreciate the relevant modifications applied to the paper.
I consider the work now more structured and justified respect the methods and theory taken into account.